# Vector Database Watermarking

Zhiwen Ren, Wei Fan, Qiyi Yao*, Jing Qiu, Weiming Zhang*, Nenghai Yu

School of Cyber Science and Technology
University of Science and Technology of China, Hefei, China

## Abstract

Vector databases support machine learning tasks using Approximate Neighbour (ANN) query functionality, making them highly valuable digital assets. However, they also face security threats like unauthorized replication. By embedding stealth information, watermarking technology can be used for ownership authentication. This paper introduces a watermarking scheme specifically designed for vector databases. The scheme consists of four steps: generating identifiers, grouping, cryptographic mapping, and modification. Since watermark embedding requires modification of certain vectors, it may negatively affect the ANN query results. Further investigation reveals that in the widely used Hierarchical Navigable Small World (HNSW) indexing structure for vector databases, heuristic edge selection and pruning strategies result in some vectors having fewer edges or even none at all. These vectors exhibit significantly lower query frequencies than others, which means that modifying these vectors incurs less impact on query results. Based on this observation, we propose the **T**ransparent **V**ector **P**riority (TVP) watermarking scheme, which prioritizes embedding the watermark in these low-query-frequency "transparent" vectors to minimize the impact of watermark embedding on query results. Experimental results show that compared to the current most effective and relevant watermarking schemes, the TVP scheme can significantly reduce the number of missed and false queries by approximately 75%.

## 1  Introduction

A vector database [25] is a data management system specifically designed to store and retrieve vector data. These vectors are generated by machine learning models and represent raw unstructured data such as text, images, and audio. By converting unstructured data into vectors, we can measure the similarity between different data by calculating the distance between the vectors. Vector databases can be applied to scenarios such as retrieval augmentation generation (RAG) [22, 16, 8], recommender systems, and similar case retrieval.

The applications of vector databases rely on approximate nearest neighbor(ANN) search. Nearest neighbor (NN) search refers to finding multiple vectors in a vector dataset that are closest to a given query vector. When the size of the vector dataset is large, the time cost of completely traversing all vectors for an exact search is too high, so the researchers proposed the approximate nearest neighbor search [34, 14, 31]. ANN algorithms can find multiple vectors that are close to the query vector in a reasonable amount of time, and even though these vectors may not be the absolute nearest vectors, they usually satisfy the practical requirements.

Criteria for evaluating ANN algorithms include recall, query response time, index construction time, space occupation, etc. After a long period of development, a variety of ANN methods have emerged, mainly including tree-based [3, 2, 37], graph-based [23, 9, 7], hash-based [32, 35, 21], and

---

*Corresponding author: {qyyao@mail., zhangwm@} ustc.edu.cn.

39th Conference on Neural Information Processing Systems (NeurIPS 2025).

quantization-based methods [15, 11, 6]. Hierarchical Navigational Small World (HNSW) stands out among many methods because of its high recall and fast query speed. Therefore, it is supported by many commercial vector database systems such as Milvus [24] and Pinecone [27], and is currently one of the most important vector database Indexing structures.

Recently, vector databases have been widely used in various AI applications. For example, in medical scenarios, medical institutions can share desensitised disease feature vectors, and patients can retrieve similar cases and the doctors or institutions that have dealt with them through similarity searches, so as to obtain medical reference suggestions without compromising the privacy of patients. This shows that vector databases have high application value, and their copyright protection issues should be given due attention. Watermarking technology [19, 10, 20, 13] is a common means of copyright protection, and watermarks can be embedded into digital assets to prove their ownership. However, existing vector data watermarking methods [33, 28] are primarily applied to coordinate vectors in Geographic Information Systems (GIS), while current database watermarking methods [**li2022secure**, 5, 12] are mainly designed for relational databases. Neither of these approaches is suitable for vector databases.

In this paper, we present for the first time a vector database watermarking scheme. The scheme is divided into four steps: identifier generation, vector grouping, cryptographic mapping and modifying partial vectors. The watermark information is embedded by modifying some vectors to change the distribution of vector cryptographic mapping results within the group. We present two vector selection strategies: random selection (RS) and transparent vector priority (TVP). RS is a basic strategy that selects vectors randomly for modification. The defect of RS is that it will change the relative positions and neighbor relationships between the vectors, and may have a negative impact on the ANN query, which is called the "embedding impact". In order to quantify this embedding impact, we define the number of missed queries and the number of false queries of the vectors before and after embedding the watermark as the evaluation metrics. In order to reduce the embedding impact of watermarking, we conduct an in-depth study on the HNSW index structure. We conjecture and confirm that due to the heuristic edge selection and pruning strategy of HNSW, some vectors have few or even no edges, and these vectors are less frequently queried compared to other vectors, and thus modifying these vectors results in few query errors. We call these vectors with low query frequency "transparent vectors".

Based on this finding, we propose a transparent vector priority vector database watermarking scheme. When selecting vectors for modification, TVP prioritizes transparent vectors to minimize the embedding effect according to the transparency threshold required by the algorithm. Experimental results show that by prioritizing transparent vectors, TVP significantly reduces the impact of watermark embedding on ANN queries while maintaining good robustness.

**Contributions**. The contributions of this paper are as follows:

- **Quantifying embedding impact**: Defining the number of missed queries and false queries to quantify the impact of watermark embedding on ANN queries for the first time.

- **Optimizing vector database watermarking schemes**: Proposing the first vector database watermarking scheme RS and minimizing embedding impact through the Transparent Vector Priority (TVP) strategy.

- **Experimental Validation**: Experimental results demonstrate that TVP significantly reduces the embedding impact and also has strong robustness.

## 2   Related Work

This section reviews existing watermarking schemes for relational databases and datasets, highlighting their development, challenges, and differences from vector database watermarking.

Agrawal et al. proposed the first relational database watermarking scheme [1], which embeds the watermark by modifying the least significant bit (LSB) of the data. Although it enables basic embedding and extraction, it is less robust and susceptible to data tampering. Subsequently, Sion et al. [30] and Cui et al. [4] improved it to enhance the robustness, but it still causes data distortion.

Reversible watermarking algorithms [5, 17, 26] have been developed to safeguard the availability of databases. These algorithms can recover the original data, but the process of recovering the

original data is also the process of removing the watermark, so the recovery privileges need to be carefully managed. Schemes based on distortion constraints limit the magnitude of data modification during watermark embedding to maintain usability, but they often require compromises in embedding capacity or extraction accuracy.

Ren et al. [29] propose a statistical property-preserving watermarking scheme for relational databases that ensures that watermark embedding does not affect the statistical properties of the data. Tabular-Mark [36], proposed by Zheng et al. makes watermarking virtually stealth to dataset applications for some machine learning tasks.

Vector database watermarking shares the same goals as these schemes, such as basic embedding/extraction, robustness and usability. However, the metrics to measure the impact of watermarking on usability are different due to different usage scenarios. Therefore, applying existing schemes directly to vector databases may not be effective.

## 3    Preliminaries

### 3.1    Approximate Nearest Neighbor (ANN)

Unstructured data (e.g., text, images, audio) can be transformed into high-dimensional vectors via feature extraction. The distance between vectors indicates the similarity of the original data. Nearest-neighbor search of vectors enables quick location of similar data in large-scale datasets, supporting intelligent applications.

Let $\mathbf{D} = \{V^1, V^2, \cdots, V^n\}$ represent a vector database with $n$ vectors, where $V^i \in \mathbb{R}^d$. For any two vectors $V_i$ and $V_j$ in $\mathbf{D}$, the Euclidean distance can be denoted as $\mu(V^i, V^j)$. Nearest neighbor (NN) search aims to find $k$ neighbor vectors $N^q = \{N_1^*, N_2^*, \cdots, N_k^*\}$ in the vector database $\mathbf{D}$ that are closest to the query vector $q$, satisfying:

$$N^q = \underset{N^q \subset D,\, |N^q| = k}{\operatorname{argmin}} \sum_{i=1}^{k} \mu(q, N_i^*) \tag{1}$$

When the dataset size is large, the time cost of NN search is high. Thus, ANN search is proposed. It relaxes the accuracy requirement and aims to find $k$ vectors $N_1, N_2, \cdots, N_k$ such that:

$$\mu(q,\ N_i)\ \leq c\ \times\ \mu(q,\ N_i^*), \quad 0 \leq i \leq k \tag{2}$$

where $c \geq 1$ is the error tolerance, meaning the distance between the found vector and the query vector is at most $c$ times the distance of the true nearest neighbor.

### 3.2    Hierarchical Navigable Small World

Hierarchical Navigable Small World (HNSW) [23] is an important graph-structured algorithm for ANN querying of high-dimensional data. It balances query efficiency and accuracy through four core ideas: hierarchical, heuristic edge selection, pruning, and greedy search.

**Hierarchical**: HNSW constructs a multi-layered small world graph. The bottom layer contains all vectors, and as the number of layers increases, the number of vectors and connected edges decreases, creating a sparse structure.

**Edge selection strategy**: When adding a vector $q$, HNSW first finds the $efConstruct$ nearest neighbours of q and checks whether $q$ should be connected to it. Given a set $S$ of vectors that have been connected to $q$, $q$ will be connected to its nearest neighbour $e$ only if $\mu(e, q) < \mu(e, o)$ of all $o$ in $S$.

**Operation strategy**: Each vector has an upper bound on the number of edges. If adding $q$ causes a vector $e$ to exceed this upper limit (set to $2M$ in the bottom layer and $M$ in the other layers), then $e$ must reselect its edges.

**Greedy search**: In HNSW for ANN search with query vector $q$ to find the $K$ closest vectors, start from the highest-level entry vector and move down layer-by-layer. At non-bottom layers, greedily find the closest vector to $q$ as the next-layer entry. At the bottom layer, maintain sets $W$ (nearest neighbors) and $C$ (candidates). Continuously pick the closest vector from $C$ to $q$ to update $W$ until $C$

is empty or no $C$ vector is closer to $q$ than $W$ vectors. When $K = 1$, this process is known as greedy search. More detailed introductions and examples can be found in Appendix A.

## 3.3 Watermarking Framework

The watermark $W \in \{0,1\}^L$ is a binary string to be embedded into digital carriers for copyright verification. A watermarking scheme for vector database $\mathcal{D}$ consists of the following two algorithms:

- **Embedding Algorithm**: $\mathrm{Em}(\mathcal{D}, W, \theta_1) \rightarrow \mathcal{D}_w$
  **Input**: the original database $\mathcal{D}$, the watermark $W$, and the embedding parameters $\theta_1$.
  **Output**: the watermarked database $\mathcal{D}_w$.
- **Extraction Algorithm**: $\mathrm{Ex}(\mathcal{D}', \theta_2) \rightarrow W'$
  **Input**: a database $\mathcal{D}'$ that may have been modified and extraction parameters $\theta_2$.
  **Output**: the extracted watermark $W'$.

Before sharing a vector database, data owners can use an embedding algorithm to embed watermarks into the database. When copyright disputes arise, an extraction algorithm can be used to extract watermarks from suspected infringing copies of the data, serving as evidence of copyright ownership.

## 3.4 Threat Model

Assume that the vector database is stolen. Then the thief may try to remove the watermark through various attacks before using the database, thus preventing us from claiming the copyright. In this context, we define the threat model for watermarking as follows:

**Full access privileges**: the attacker has all the privileges of the vector database and can perform add, delete, change, and query operations.

**Knowledge of the algorithm but not the parameters**: the attacker is familiar with the embedding algorithm of the watermark, but does not know the specific parameters used for embedding.

**The intensity of attacks is constrained**: Since attackers themselves are also database users, they must maintain retrieval functionality while attempting to remove watermarks. Therefore, they will only apply attacks of limited intensity to avoid a significant decline in retrieval accuracy.

# 4 Basic Methodology: Random Selection

This section describes a basic watermarking scheme for vector databases, which generates a unique identifier $ID$ for each vector, divides the vectors into groups based on the identifier, maps each vector as '1-vector' or '0-vector', and adjusts the ratio of different types of vectors in each group by modifying the vectors to embed the watermark information. Since the vectors are randomly selected for modification, the scheme is called a Random Selection Scheme (RS). In the following, we describe the watermark embedding and extraction process in detail.

## 4.1 Watermark Embedding

The watermark embedding process consists of four steps: generating vector identifier, vector grouping, cryptographic mapping and vector modification. The pseudo-code is shown in Algorithm 1 in Appendix B.

**Generate vector identifier**: To avoid relying on the original vector identifiers, we use the Most Significant Bit (MSB) method [36] to generate identifiers directly from the vector data. Initialise the random number generator using the seed parameter to generate a set of random dimension indices $\mathcal{I} = \{d_1, d_2, \ldots, d_m\}$. Extract the highest bits from each selected dimension and concatenate these bits to form an identifier $ID_V$:

$$ID_V = M(V_{d_1})M(V_{d_2})\ldots M(V_{d_m}) \tag{3}$$

where $M(x)$ denotes the highest valid bit of $x$. For example, given a vector $V = (42, 23, 78, 14)$ and a randomly chosen dimension $\mathcal{I} = \{2, 3\}$, the identifier is $M(78)M(14) = 71$.

**Vectors grouping**: Watermark $W$ are binary sequences defined by the database owner. If the length of the watermark is $L$, the vectors are partitioned into $L$ non-overlapping groups. A vector $V$ is assigned to group $g$ when the result of taking the modulo of the hash value of its identifier $(ID_V)$ with $L$ is equal to $g$, that is, $H(ID_V) \mod L = g$.

**Cryptographic mapping**: each vector $V$ is converted to a binary value (0 or 1). First, the target embedding dimension $d_w = H(ID_V) \mod d$ is computed. To avoid changing the identifier $ID_V$ of the vector during the watermarking process, if $d_w$ is in the dimensions set $\mathcal{I}$ used to generate $ID_V$, it is iteratively updated as $d_w = (d_w + 1) \mod d$ until $d_w$ is outside of that set. Next, the value $V_{d_w}$ is extracted from the target dimension, converted to a binary $b$ of length $l$, and the index $i = H(ID_V) \mod (l \times f) + 1$ $(0 < f \leq 1)$ is computed. The $(l-i)$th bit of $b$ is used as the initial mapping result. The factor $f$ $(0 < f \leq 1)$ can modulate the size of the index $i$, indirectly affecting the magnitude of potential modifications in subsequent processes. For example, for the vector $V = (42, 23, 78, 14)$, if $d_w = 3$ conflicts with the dimension that produces $ID_V$, it is updated to $(3+1) \mod 4 = 0$. With $V_0 = 42$ (in binary form $b = 101010$, $l = 6$) and $i = 3$, the initial mapping bit is $b[3] = 0$.

However, this approach faces the "sparsity curse": sparse vectors in the database tend to map primarily to 0, which skews the distribution and weakens the effectiveness of the watermark. To alleviate this problem, we introduce an enhancement strategy: XOR the initial result $w$ with the least significant bit of $(l-i)$, i.e., $w = w \oplus ((l-i) \mod 2)$. The pseudo-code is shown in Algorithm 2 in Appendix B.

**Vector modification**: Vectors mapped to 1 or 0 are respectively referred to as "1-vector" or "0-vector". To embed the $g$-th bit $W_g$ of the watermark $W$ into the $g$-th group of vectors, the proportion of vectors that can be mapped to the value of $W_g$ needs to be increased to the preset watermark strength $s$. For example, if $W_g = 1$ and the $g$-th group contains 1,000 vectors, among which 500 are 1-vector, when $s = 0.7$, it means that we expect the proportion of 1-vector to be 0.7, that is, there should be 700 1-vector. In this case, 200 0-vector need to be converted into 1-vector.

Calculate the number of vectors $n_m$ that need to be modified for each group, and then randomly select $n_m$ vectors from these vectors that cannot be cryptographically mapped to $W_g$ as the carrier vector $C_V$. Change their cryptographic mapping results by flipping the bit $b[l-i]$. For instance, given $b = 101010$ and $i = 3$, flipping $b[3]$ will change the mapping from $b[l-i] \oplus ((l-i) \mod 2) = 1 \oplus 1 = 0$ to $0 \oplus 1 = 1$. By traversing all groups and adjusting the distribution of vectors, the watermark embedding can be completed.

## 4.2 Watermark Extraction

The extraction process uses the same seed $seed$ as the embedding to ensure that the vector groupings and cryptographic mappings are the same. For each group $g$, the watermark bit $W_g$ is determined by majority voting: if the number of 1-vectors exceeds the number of 0-vectors, then $W_g = 1$; otherwise, $W_g = 0$. Iterate over all groups to reconstruct the extracted watermark $W'$. Pseudo-code is provided in the Algorithm 3 in Appendix B.

## 5 Embedding Impact

In this section, we define and model the impact of watermark embedding on the query process to guide us in optimizing the watermarking approach.

To quantitatively measure the negative impact of watermarking on vector ANN queries, we give the following definition:

**Definition 1**: (Nearest Neighbor Response Domain). The nearest neighbor response domain $R_V$ of a vector $V$ is defined as:

$$R_V \triangleq \{q | q \in \mathbf{D}, V \in N^q\}. \tag{4}$$

For a vector $V \in \mathbf{D}$, all the query vectors $q$ from $\mathbf{D}$ that consider $V$ as a nearest neighbor vector make up the nearest neighbor response domain $R_V$ of $V$.

**Definition 2**: (Number of Missed Queries). The number of missed queries $M_V$ of vector $V$ is defined as:

$$M_V \triangleq |R_V \setminus R_{V_W}|, \tag{5}$$

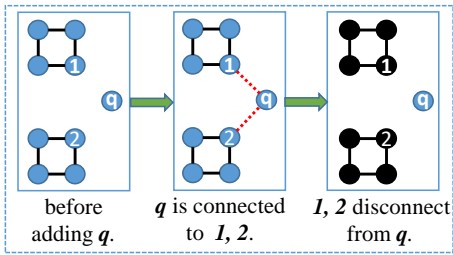 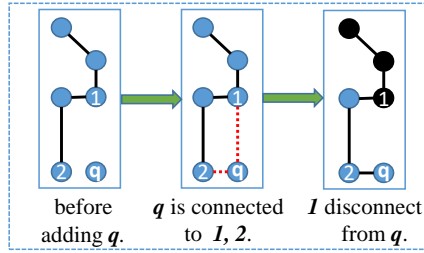

| before | $q$ is connected | $1, 2$ disconnect | | before | $q$ is connected | $1$ disconnect |
| adding $q$. | to $1, 2$. | from $q$. | | adding $q$. | to $1, 2$. | from $q$. |

(a) Vector $q$ has no edge.      (b) The query is stuck in a local optimum.

Figure 1: Two cases in which the vector $q$ cannot be queried.

where $\setminus$ represents the relative complement operation and $A \setminus B$ is the set of elements in $A$ but not in $B$. $V$ and $V_W$ respectively denote the vector data before and after the watermark embedding. Thus, the number of missed queries for $V$ refers to the number of vectors that originally belonged to the nearest neighbor response domain $R_V$ of $V$ but are absent from the nearest neighbor response domain $R_{V_W}$ of $V_W$.

**Definition 3**: (Number of False Queries). The number of false queries $F_V$ of vector $V$ is defined as:

$$F_V \triangleq |R_{V_W} \setminus R_V|. \tag{6}$$

The Number of false queries of $V$ refers to the number of vectors that belong to the nearest neighbor response domain $R_{V_W}$ of $V_W$ but are missing in the nearest neighbor response domain $R_V$ of $V$.

The selection of carrier vectors $C_V$ should be considered from the following two aspects:

**Low embedding impact**: in order to minimize the impact of watermarking on query results, i.e., to reduce $M_V$ and $F_V$, vectors that are rarely queried should be preferred as carrier vectors.

**High robustness**: in order to prevent the attacker from accurately locating and deleting the carrier vectors by their features, the selection of vectors with too obvious features should be avoided.

# 6 Methodology: Transparent Vector Priority

This section introduces two key concepts, **Stealth vectors** and **Transparent vectors**, which are the basis of our proposed scheme. Based on the study of these two types of vectors, a **T**ransparent **V**ector **P**riority scheme is proposed to reduce the impact of embedded watermarks on query results.

## 6.1 Stealth Vectors and Transparent Vectors

**Stealth Vectors**: We conjecture that the heuristic edge selection and pruning strategy of HNSW may produce edgeless vectors. Take Fig. 1(a) as an example, the vector $q$ is connected with vector 1 and 2 such that the number of edges of vector 1 and 2 increases to three. If the maximum number of edges is set to 2, the vector 1 and 2 need to be re-selected according to the pruning strategy. Since they already have two neighbours closer than $q$, they will not be connected to $q$, and $q$ becomes an edgeless vector. Experiments on the **ANN_SIFT1M** dataset verified the existence of such vectors, and the detailed procedure is shown in Appendix C; only the conclusions are presented here. With parameters $M = 8$ and $efConstruct = 100$, the constructed HNSW graph contains 12 connected branches, 11 of which contain only one edgeless vector. In further experiments, all vectors are used as query vectors to retrieve the 10 nearest neighbours, and the number of times each vector is queried is recorded, which shows that the edgeless vector is never queried.

In addition, 40 vectors in the maximum connected branch are never queried, presumably due to the search algorithm falling into a local optimum. As shown in Fig. 1(b), vector 1 is disconnected from the edge of $q$ due to the pruning strategy, and the greedy algorithm will mistakenly take vector 1 as the result when querying $q$. Experiments show that the average degree of the unqueriable vectors in the maximal connected branch is 1.87, while the average degree of the random 1000 vectors is 8.92, which verifies the property that such vectors have fewer edges.

Based on the above characteristics, we define these two types of unqueriable vectors as Stealth vectors.

**Transparent Vectors**: Since stealth vectors are never queried, embedding watermarks in such vectors has minimal impact on queries. However, stealth vectors are not optimal watermarking vectors, as they are recognised by their distinctive features, and there is a risk of removing the watermark by deleting all non-queryable vectors. To balance the transparency and robustness of watermarking, we prefer to use vectors that are queried less frequently as carrier vectors, calling such vectors transparent vectors.

Based on the feature that stealth vectors are "far away from other vectors and have fewer edges", we propose the hypothesis that vectors with longer edges and fewer edges are more likely to be transparent vectors. We test this hypothesis experimentally. The experiments found that the number of times a vector is queried decreases when the average edge length of the vector increases, while a decrease in the number of edges also leads to a decrease in the number of queries.

To quantitatively measure the number of times a vector is queried, we construct the transparency parameter $ts$ by combining the average edge length $l$ and the number of edges $e$ of the vector:

$$ts \triangleq \frac{l - m_l}{\delta_l} - \frac{e - m_e}{\delta_e}, \tag{7}$$

where $m_l$ and $\delta_l$ denote the mean and standard deviation of the variable $l$. To reduce the computational overhead, some of the vectors can be sampled for computation.

Table 1: Correlation coefficient between different parameters and the count of queries.

| parameter | $ts$ | $l$ | $e$ |
|-----------|------|-----|-----|
| $\rho$ | **-0.6998** | -0.5565 | 0.5131 |

To compare the correlation $e$ between the transparency parameter $ts$, the average edge length $l$ and the number of edges with the number of queries, the correlation coefficients $\rho$ between them and the number of queries were calculated, and the results are shown in Table 1. It can be seen that the absolute value of the correlation coefficient between the transparency parameter $ts$ and the number of queries is the largest, which indicates that $ts$ has the strongest correlation with the number of queries, and therefore $ts$ is more suitable to be used as a parameter to measure the transparency of the vector. Furthermore, we calculated the Pearson correlation coefficient between the transparency parameter $ts$ and query frequency under various HNSW configurations (see Appendix C, Table 3). The results consistently fell below -0.66, indicating that $ts$ exhibits a stable and consistent negative correlation with the query frequency, thereby serving as a reliable transparency metric.

In summary, we confirm the existence of transparent vectors in HNSW and conclude that: due to the edge pruning strategy, some vectors far away from other vectors have few edges or even no edges, which makes them difficult to be retrieved; the transparency of vectors can be measured by the transparency score $ts$, and the larger $ts$ is, the more likely it is a highly transparent vector.

## 6.2 Transparent Vector Priority (TVP)

This section introduces the Transparent Vector Priority (TVP) vector database watermarking scheme.TVP follows the same framework as the RS scheme, but prioritises highly transparent vectors as carrier vectors to minimise query errors caused by watermark embedding.

Taking advantage of the edge correlation property of HNSW, we use transparent vectors with low query frequency as the ideal carrier. Theoretically, in order to find these transparent vectors accurately, it is necessary to count the number of queries for each vector through a large number of queries, from which the vectors with low query frequency are filtered out. Based on the study of transparent vectors in the previous section, we can quickly find transparent vectors by using the $ts$ parameter.

Specifically, the TVP algorithm uses the threshold parameter $\tau$ (0 - 1) to select the first $\tau \times 100\%$ of transparent vectors. It randomly samples the vectors, computes the mean and variance of their edge length $l$ and number of edges $e$, computes the transparency score $ts$, and determines the filtering threshold $ts_\tau$ based on the top $\tau$ ranked vectors.

When the embedding algorithm needs to select vectors for modification, it no longer selects vectors randomly, but prefers vectors with transparency score ts greater than the threshold $ts_\tau$, i.e., vectors with higher transparency and lower query frequency. If $\tau$ is set smaller, there may not be enough eligible vectors, and the algorithm will only select vectors with slightly lower transparency scores.

# 7 Experiments

A watermarking algorithm can be evaluated from two aspects: embedding impact and watermark robustness.

We use the average number of missed queries ($AMQ$) and the average number of false queries ($AFQ$) of watermarked vectors containing watermarks after adding watermarks to measure the embedding impact. The specific calculation methods of these two indicators are as follows:

$$\mathbf{AMQ} :\triangleq \frac{\sum_{V \in C_V} M_V}{|C_V|}, \mathbf{AFQ} :\triangleq \frac{\sum_{V \in C_V} F_V}{|C_V|}. \tag{8}$$

Watermark robustness is the ability to accurately extract watermarks from watermark-containing vectors under attack. Given the threat model in section 3.4, an attacker who knows about our watermarking approach will use an adaptive attack strategy. All robustness experiments involve such adaptive attacks. Among them, adaptive deletion and modification attacks are the most effective. In the former attack, the attacker removes a certain percentage of transparent vectors to remove the watermark. In the latter attack, the attacker modifies the values of some of the dimensions of the partially transparent vector. We measure robustness using the bit error rate (BER) between the extracted watermark and the embedded watermark, which is calculated as:

$$\mathbf{BER} :\triangleq \frac{\sum_{i=1}^{L} w_i \oplus w_i^{'}}{L}, \tag{9}$$

where $w_i$ is the $i$-th watermark, and $w_i^{'}$ is the extracted $i$-th watermark.

The data and environment used for the experiments are consistent with Section 6.1, i.e., the commonly used vector dataset ANN_SIFT1M[18] was implemented and experimented with using Python and Faiss libraries on PCs equipped with AMD Ryzen 5 5600G processors. Experimental results are generally the average of multiple runs. The default parameters for the experiment are $M = 8$, $efConstruct = 100$, and $k = 100$ neighbors per query.

## 7.1 Comparison

In this section, we apply state-of-the-art relevant watermarking schemes to vector databases and compare them with TVP. Specifically, they include SCPW, a watermarking scheme for relational databases proposed in 2023; TabularMark, a watermarking scheme for tabular datasets proposed in 2024; and RS, a scheme proposed in Section 4 of this paper. In the comparison experiments, the public parameters of all the schemes are kept the same, while the private parameters of the schemes are chosen to have the optimal configurations according to their characteristics.

Table 2: Embedded impacts of different schemes.

| Scheme | SCPW[29] | TabularMark[36] | RS | TVP |
|---|---|---|---|---|
| AMQ | 14.26 | 12.76 | 13.52 | 3.15 |
| AFQ | 14.67 | 13.15 | 13.65 | 3.21 |

Table 2 demonstrates the AMQ and AFQ before and after embedding watermarks using different schemes. The results show that the AMQ and AFQ of the TVP scheme are reduced by approximately 75% compared to the other schemes. This is because the other schemes do not fully consider minimizing the impact of watermarking on queries, while TVP reduces the number of false and missed queries by preferring transparent vectors that have been queried less often to embed watermarks.

Additionally, we evaluated the variance and maximum values of missed queries (MQ) and false queries (FQ) across different schemes. The results indicate that TVP exhibits lower volatility. Detailed findings are presented in Appendix D Table 4. We further evaluated generalization capabilities of TVP on other datasets, with results indicating it consistently achieves significant reductions in AMQ and AFQ across all datasets. Detailed results are presented in Tables 5 and 6 in Appendix D.

Since TabularMark does not embed watermarked plaintext, its robustness cannot be directly measured using BER. Therefore, following the method proposed by the authors of the scheme, we use the

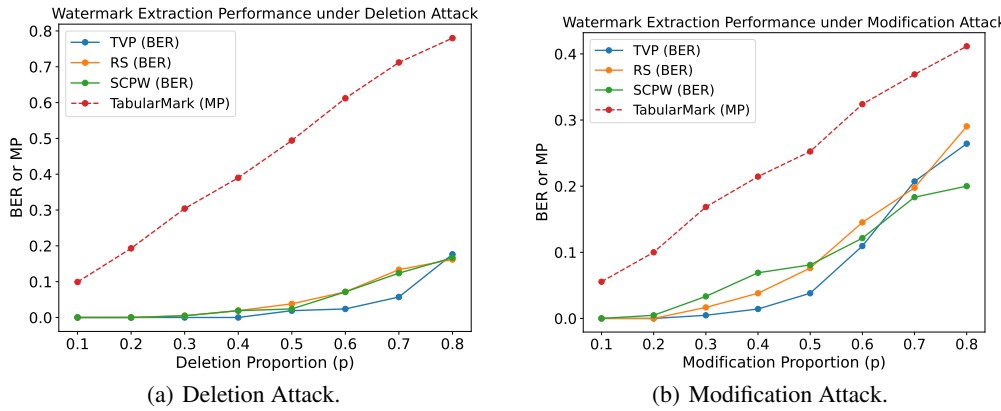

(a) Deletion Attack.         (b) Modification Attack.

Figure 2: Watermark robustness of different schemes.

proportion of watermarks flipped due to vector modification, i.e., Mismatch Percentage (MP), as a metric for robustness under modification attack. For deletion attack, the proportion of watermarked vectors deleted is used as the robustness metric. Since deletion also leads to mismatches, we also denote this metric by MP.

Figure 2(a) illustrates the BER or MP of different schemes after being subjected to a deletion attack. Different colored lines represent different schemes. It can be seen that the $BER$ of TVP is comparable to other schemes, indicating that its robustness can be as strong as existing schemes. Figure 2(b) shows the BER or mismatch percentage of different schemes after a modification attack. Again, the robustness of TVP is similar to the existing schemes.

In summary, the experimental results show that TVP not only significantly reduces query errors but also maintains considerable robustness compared to the current optimal watermarking scheme. Therefore, TVP is currently the most suitable watermarking scheme for vector databases.

## 7.2 Method Validation And Parameter Analysis Experiments

This section presents a more comprehensive set of experiments designed to: (1) test the effectiveness of the proposed TVP strategy; (2) investigate the impact of key parameters on embedding impact and watermark robustness, thereby providing guidance for parameter selection. Detailed experimental procedures and most of the results are presented in the Appendix E.

**TVP Strategy Effectiveness**: We evaluated the impact of TVP strategy on query performance under different HNSW parameter configurations. Results indicate that both AMQ and AFQ remain at low levels and are minimally affected by parameter variations. Detailed results are presented in Appendix E Table 7. To evaluate the impact of TVP on recall rate, we compared the recall rate (R@k) at different k values before and after watermark embedding. As shown in Appendix E Table 8, the recall rate remained virtually unchanged, indicating that TVP has a negligible effect on retrieval accuracy. We evaluated the generalization capability of the TVP strategy across multiple ANN index structures. By recording the query frequency of vectors, TVP prioritises embedding watermarks in low-frequency vectors. This strategy was compared against the baseline method of randomly selecting vectors (RS). Experimental results, as shown in Appendix E Table 9, demonstrate that TVP reduces the impact on query performance caused by embedding across all indexes. Therefore, the concept of transparent vector priority applies not only to HNSW but also to other ANN indexes.

**Embedding Impact**: The parameter $\tau$ significantly influences both AMQ and AFQ. A smaller value of $\tau$ means that the algorithm prefers more transparent vectors as the carrier vectors, which drastically reduces the number of false queries and missed queries. As shown in Fig. 7 of Appendix E, the smaller $\tau$ is, the less impact it has on the query. The parameter $s$ affects the number of vectors that need to be modified. The larger the $s$, the more vectors need to be adjusted to match the watermark bits, which has a greater impact on the query result.

**Watermark Robustness**: The larger value of $s$ enables more carrier vectors in each group after embedding to be mapped to the target watermarked bits and enhances the robustness of watermarking. Fixing other parameters, experiments are conducted on different combinations of $s$ and deletion

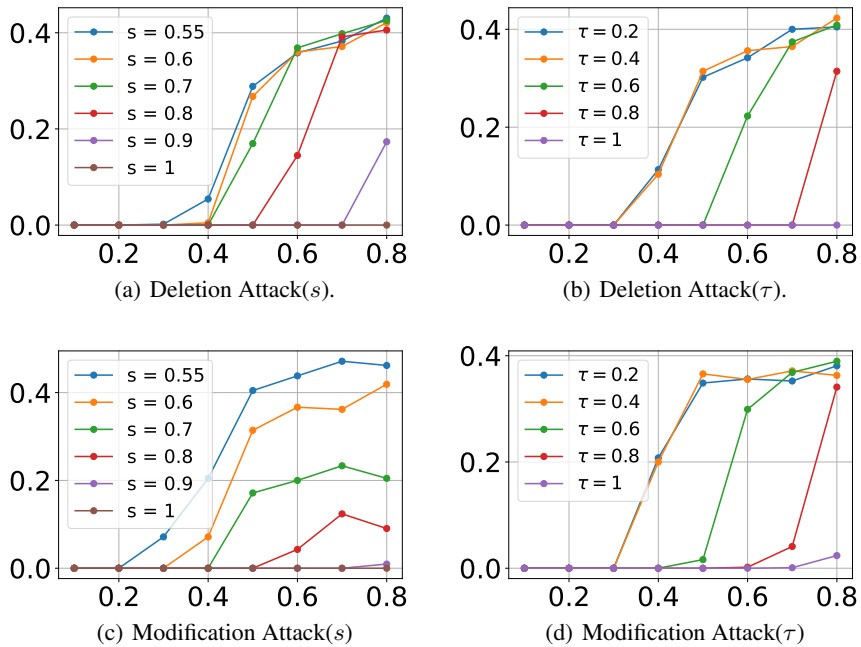

(a) Deletion Attack($s$).      (b) Deletion Attack($\tau$).

(c) Modification Attack($s$)      (d) Modification Attack($\tau$)

Figure 3: The figure illustrates the impact of parameters on watermark robustness. The horizontal axis represents the attack intensity $p$, while the vertical axis shows the **BER** (Bit Error Rate).

attack proportion $p$, and the results are shown in Fig. 3(a). It can be seen that under the same attack strength, the larger the value of $s$, the smaller the BER, i.e., the stronger the robustness. The results of the modified attack experiments (Fig. 3(b)) also confirm this conclusion. The smaller the $\tau$ value is, the more prominent the carrier vector features are, and the watermark is easily damaged by the adaptive attack. From Fig. 3(c) and Fig. 3(d), the smaller $\tau$ is, the larger BER is, and the weaker the robustness is.

# 8 Conclusion

This paper explores watermarking techniques in vector databases and proposes the first watermarking scheme, RS. However, RS will modify some vectors, leading to false queries and missed queries in ANN queries. To reduce these embedding impact, we conducted an in-depth study of the most popular index structure, HNSW, and discovered its characteristic: certain vectors in the HNSW graph have few or no edges, resulting in lower query frequencies for these vectors. Modifying these "transparency vectors" will not result in too many false queries and missed queries. Building upon this insight, we refine RS into TVP, prioritising transparent vectors as watermark carriers to minimise embedding impact. Experimental results demonstrate that TVP significantly reduces false queries and missed queries while exhibiting robust performance.

**Limitations**: This paper presents a transparency score $ts$ construction formula specifically for HNSW. For other index structures, no suitable $ts$ currently exists to enable rapid selection of transparent vectors. In future work, we shall explore concrete methods for extending the concept of transparent vectors priority to other index structures.

# 9 Acknowledgements

The authors thank the anonymous reviewers and meta-reviewers for their time and constructive feedback. This work was supported in part by the Natural Science Foundation of China under Grant U2336206, 62102386, 62372423, and 62121002.

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

# A Hierarchical Navigable Small World

Hierarchical Navigable Small World (HNSW)[23] is an efficient graph-structured algorithm for ANN querying, aiming at accelerating the querying process of high-dimensional data, and it is the most dominant vector database index. The core ideas of HNSW include hierarchization, heuristic edge selection strategy, pruning strategy and greedy search. The query efficiency and result accuracy are balanced by quickly locating the query target in the high-level graph, and then searching in a refined way through the low-level graph.

**Hierarchization**: HNSW constructs a small-world graph containing multiple layers. At the bottom layer, the graph contains all vectors and constitutes the edges of the graph by connecting each point to several vectors in its neighborhood. As the number of layers of the graph increases, the number of vectors in each layer decreases, and the number of connected edges also decreases, creating a sparse graph structure.

**Heuristic edge selection strategy**: At each layer, HNSW uses a heuristic edge selection strategy to determine which vectors the newly added vector should be connected to. Let $efConstruct \in \mathbb{N}$ be a parameter, that for a vector $q$ to be added to HNSW, the $efConstruct$ near-neighbors of $q$ are first found and then traversed in order of proximity to $q$. Here it is important to distinguish between the concepts of near-neighbor and neighbor; for a vector, a near-neighbor is a vector that is close to it, and a neighbor is a vector that is connected to it and can be accessed by each other. The set of vectors already connected to $q$ is denoted as $S$. For a near-neighbor $e$ of $q$, $q$ will only connect to $e$ if the distance from $e$ to $q$ is less than the distance from $e$ to any vector in $S$ (i.e., $\forall o \in S, \mu(e, q) < \mu(e, o)$).

**Pruning strategy**: Let $M \in \mathbb{N}$ be a parameter that controls the upper limit of the number of edges for each vector. If the addition of $q$ causes the vector $e$ to have more edges than the upper limit, $e$ must re-select its edges to conform to the edge limit. This upper limit is set to $2M$ in the bottom layer of the graph, and $M$ in the other layers.

For example, set $efConstruct = 3$, $M = 1$. As shown in Figure 4(a), there are already three vectors 1, 2, and 3 in the HNSW graph, with two edges $E(1, 2)$ and $E(1, 3)$. Now add a vector $q$, where $\mu(1, q) < \mu(2, q) < \mu(3, q)$. Follow these steps to determine whether $q$ connects to 1, 2, and 3: First, connect to 1, since $q$ has no neighbors yet; do not connect to 2, because $\mu(2, q) > \mu(2, 1)$; connect to 3, because $\mu(3, q) < \mu(3, 1)$. At this point, the heuristic edge selection for vector $q$ is complete, as shown in Figure 4(d); due to the addition of $q$, the number of edges for vector 1 increases to 3, exceeding the upper limit $2M = 2$, so vector 1 needs to prune some edges. The final HNSW graph is shown in Figure 4(f).

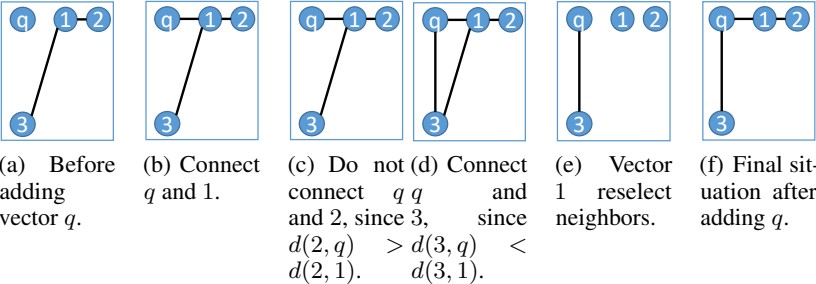

(a) Before adding vector $q$.

(b) Connect $q$ and 1.

(c) Do not connect $q$ and 2, since $d(2, q) > d(2, 1)$.

(d) Connect $q$ and 3, since $d(3, q) < d(3, 1)$.

(e) Vector 1 reselect neighbors.

(f) Final situation after adding $q$.

Figure 4: Add a new vector $q$ to HNSW using heuristic edge selection and pruning strategies.

**Greedy search**: The ANN search process in HNSW is described next. Suppose we have a query vector $q$, and the goal is to find the $K$ closest vectors to $q$. The search process starts at the highest level of the entry vector of the HNSW and proceeds downward layer by layer. At each layer except the bottom layer, a greedy search is performed to find the closest vector to the query vector $q$ at that layer, which serves as the entry vector for the next layer. At the bottom layer, two sets $W$ (the set of nearest neighbor vectors) and $C$ (the set of candidate vectors) need to be maintained, and the closest vector to $q$ from $C$ is continuously selected to update $W$ until $C$ is empty or the vectors in $C$ are no longer closer to $q$ than any vectors in $W$. At this point the vectors kept in $W$ are the nearest neighbors of $q$. When $K = 1$, this process is known as greedy search.

## B  Pseudo-code

---

**Algorithm 1:** Watermark Embedding

---

**Input:** Vector database $D$, Watermark $W$, Strength $s$, Random seed $seed$, Limiting factor $f$

**Output:** Watermarked vector database $D_W$

1   $d \leftarrow$ dimension of a vector in $D$

2   $L \leftarrow$ length of the watermark $W$

3   Initialize $L$ empty groups $G$

4   Initialize random number generator with $seed$

5   Randomly generate the set of positive integers $\mathcal{I}$

6   **for** $V \in D$ **do**

7      $ID_V \leftarrow$ the identifier of $V$

8      $g \leftarrow H(ID_V) \mod L$

9      $G_g \leftarrow G_g \cup V$

10     $w \leftarrow$ **Cryptographic Mapping**$(V, \mathcal{I}, f, d)$

11     $n_{gw} \leftarrow n_{gw} + 1$ // $n_{g1}$ and $n_{g0}$ represent the number of 1-vector and
        0-vector in group $g$, respectively, with initial value 0.

12   **end**

13   **for** $g \leftarrow 0$ *to* $L - 1$ **do**

14     **if** $W_g = 1$
      // $W_g$ is the $g$-th position of the watermark.

15      **then**

16        $n_m \leftarrow max(0, s * |G_g| - n_{g1})$
        // Calculate the number of vectors $n_m$ that need to be modified .

17        $C_V \leftarrow$ randomly select $n_m$ 0-vector for modification

18      **else**

19        $n_m \leftarrow max(0, s * |G_g| - n_{g0})$

20        $C_V \leftarrow$ randomly select $n_m$ 1-vector for modification

21      **end**

22     **for** $V \in C_V$ **do**

23       $V_W \leftarrow$ **Vector Modification**$(V, \mathcal{I}, f, d)$

24     **end**

25   **end**

---

 

---

**Algorithm 2:** Cryptographic Mapping

---

**Input:** Vector $V$, Set $\mathcal{I}$, Limiting factor $f$, Dimension $d$

**Output:** Mapping bit $w$

1   $d_w \leftarrow H(ID_V) \mod d$

2   **while** $d_w \in \mathcal{I}$ **do**

3     $d_w \leftarrow (d_w + 1) \mod d$

4   **end**

5   $b \leftarrow$ the binary form of $V_{d_w}$

6   $l \leftarrow$ the length of $b$

7   $i \leftarrow H(ID_V) \mod (l \times f) + 1$

8   $w \leftarrow b[l - i] \oplus ((l - i) \mod 2)$

---

**Algorithm 3:** Watermark Extraction

**Input:** Watermarked Database $D_W$, Watermark Length $L$, Random Seed $seed$
**Output:** Extracted Watermark $W'$

1  $d \leftarrow$ dimensions of a vector in $D_W$
2  Initialize random number generator with $seed$
3  Randomly generate the set of positive integers $\mathcal{I}$
4  **for** $V \in D$ **do**
5      $ID_V \leftarrow$ the identifier of $V$
6      $g \leftarrow H(ID_V) \mod L$
7      $w \leftarrow$ **Cryptographic Mapping**$(V, \mathcal{I}, f)$
8      $n_{gw} \leftarrow n_{gw} + 1$
9  **end**
10 **for** $g \leftarrow 0$ *to* $L - 1$ **do**
11     **if** $n_{g1} \geq n_{g0}$ **then**
12         $W_g \leftarrow 1$
13     **else**
14         $W_g \leftarrow 0$
15     **end**
16 **end**
17 $W' \leftarrow W_0 || W_1 || \ldots || W_{L-1}$

## C  Stealth Vectors and Transparent Vectors

### C.1  Stealth Vectors

According to the heuristic edge selection and pruning strategy of HNSW, we believe that there are vectors with no edges in the bottom layer, and such vectors are difficult to query. As shown in Figure 5(a), vector $q$ is chosen to be connected to vectors 1 and 2, resulting in these two vectors having three edges both. If the upper limit of the number of edges is set to 2, according to the pruning strategy, vectors 1 and 2 will reselect edges. If vector $q$ is far away from them, they will not connect to $q$ as they already have two neighbors that are closer than $q$. Since $q$ does not have any edges, it cannot be queried from any other vector as an entry vector.

To test whether such vectors exist, we employ the widely-used vector dataset **ANN_SIFT1M** [18], as an example. The experiment was conducted on a PC equipped with an AMD Ryzen 5 5600G processor and implemented using Python and the Faiss library. Specifically, we test whether the graph at the bottom of the HNSW is divided into multiple connected components. When $M = 8$ and $efConstruct = 100$, there are 12 connected components in the graph constructed by HNSW, yet 11 of them contain only one vector without edges. All the other vectors are in the last biggest connected. This confirms the existence of vectors without edges.

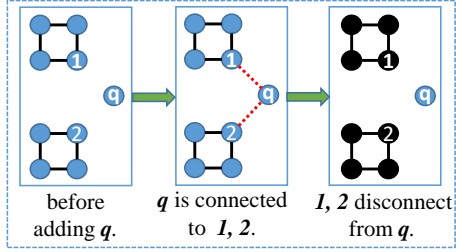

before adding *q*.    *q* is connected to *1, 2*.    *1, 2* disconnect from *q*.

(a) Vector $q$ has no edge.

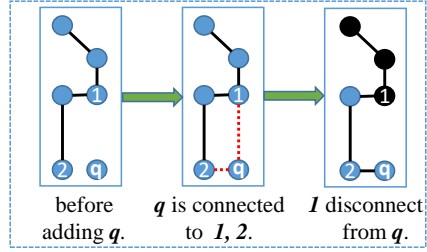

before adding *q*.    *q* is connected to *1, 2*.    *1* disconnect from *q*.

(b) The query is stuck in a local optimum.

Figure 5: Two cases in which the vector $q$ cannot be queried.

Next we test whether these vectors are unsearchable. We queried the 10 nearest neighbors using all vectors as query vectors and recorded the number of times each vector was queried. The edgeless vectors are never queried, which is as expected. However, we also find that 40 vectors in the biggest connected component are never queried. A possible reason is that the greedy algorithm used in

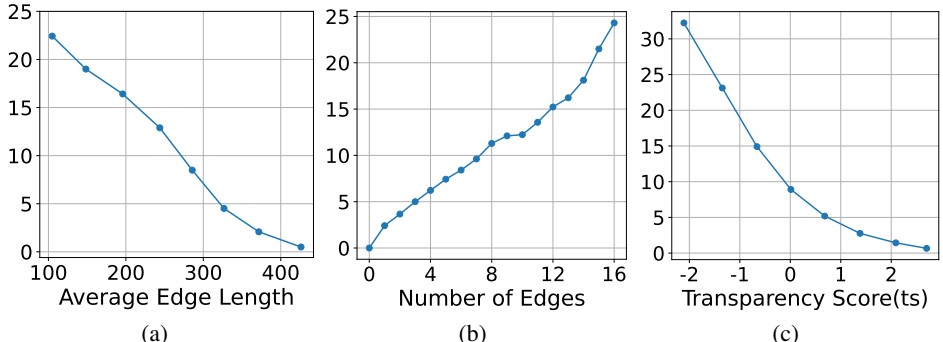

Figure 6: Experimental analysis of the effect of vector parameters on transparency. The vertical axis represents the average number of times queried.

the search process falls into a local optimum, resulting in another class of unsearchable vectors that are difficult to search for. As shown in Figure 5(b), due to the pruning strategy of HNSW, $q$ would only have neighbor 2. When querying vector $q$, if the current vector is 1, the algorithm will consider 1 to be the query result and fail to find the real search result $q$ since for any neighbor $v$ of 1, $\mu(v, q) > \mu(1, q)$. From the fact that $q$ cannot be queried starting from all black nodes in the graph. This explains why there are vectors in the main connected component that are not queried. It can be seen that the edge between $q$ and 1 is the key edge that determines whether $q$ can be queried or not, and when the key edge is pruned, $q$ becomes difficult to be queried.

To summarize, the above properties of unsearchable vectors lead to a possible conclusion: vectors with fewer edges are more likely to be unsearchable due to the prune of key edges. Indeed, in the above test, the average degree of vectors that are never queried in the main connected component is 1.87, while the average degree of 1000 randomly chosen vectors is 8.92, which confirms this conclusion to some extent. We define stealth vectors as follows.

**Definition 4**: (Stealth Vector). A vector $V \in \boldsymbol{D}$ is a stealth vector if

$$R_V = \varnothing. \tag{10}$$

Stealth vectors are indeed vectors whose nearest neighbor response domain is empty. According to the above analysis, there are two classes of stealth vectors: the first category is the no-edge vectors, which are far away from other vectors, and the pruning strategy causes all neighboring vectors to be disconnected from them; and the second category is the vectors that have a small number of edges, but their key edges are disconnected due to the pruning strategy, which makes these vectors unsearchable when searched using the greedy algorithm.

### C.2 Transparent Vectors

Because stealth vectors are never queried, embedding watermarks in such vectors has the lowest embedding impact. However, stealth vectors are not the most suitable candidates for watermarking since their distinct characteristics make them easily identifiable, resulting in the risk of watermark remove by deleting all vectors that cannot be queried. To better balance the watermark transparency and robustness, we expand the range of carrier vectors available for selection with relatively low transparency requirements. These vectors are named transparency vectors with the following definition:

**Definition 5**: (Transparency Vector). A vector $V \in \boldsymbol{D}$ is a $t-$transparency vector if

$$|R_V| \leq t. \tag{11}$$

The transparency of a vector is measured using the size of the nearest neighbor response domain; the smaller the nearest neighbor response domain is, the higher transparent the vector is. The Stealth vector is an extreme case of the transparent vector when $t = 0$.

Longer edges and fewer edges characterize stealth vectors, thus we have the following hypothesis: Vectors with longer and fewer edges are likely to be transparent vectors. To validate this, a series of validation experiments were conducted. We take the average edge length of the vector at the bottom

layer of the graph as its average edge length. In order to examine the effect of the average edge length on the number of queries, we divide the range of average edge lengths and group the vectors within the same range into one class, and calculate the average number of queries for each class of vectors. The experimental results are shown in Fig. 6(a), where the horizontal axis represents the average edge length and the vertical axis represents the average number of queries. It can be seen that the number of queries decreases with the increase of the average edge length, especially when the average edge length is extremely large, the number of queries is close to zero, indicating that the long edge vectors are rarely queried. Similarly, we verified the effect of the number of edges of a vector on the number of queries. We calculated the average number of queries for vectors with the same number of edges. The experimental results are shown in Fig. 6(b), which shows that as the number of edges decreases, the average number of queries for the vector decreases.

The transparency parameter $ts$ is constructed from the vector's average edge length $l$ and the number of edges $e$, serving to evaluate the frequency with which the vector is queried. The specific formula for calculating $ts$ is provided in Equation( 7) of the main text. A larger $ts$ indicates either a greater average edge length or fewer edges in the vector, and such vectors are typically queried less frequently. Based on this, it can be hypothesised that the transparency parameter $ts$ exhibits a negative correlation with the number of queries. To validate this hypothesis, we experimentally analysed the relationship between $ts$ and query frequency. Specifically, we divided the range of $ts$ into several intervals, grouped vectors within each interval into a category, and calculated the average query frequency for each category. The experimental results are shown in Figure 6(c), where the horizontal axis represents $ts$ and the vertical axis denotes the average query count. It can be observed that as $ts$ increases, the average query count decreases, confirming the hypothesis of a negative correlation between $ts$ and query frequency. Concurrently, Table 1 in the main text lists the correlation coefficients between the average edge length $l$, the number of edges $e$, and $ts$ with the number of queries. The results indicate that $ts$ exhibits the highest absolute correlation coefficient with the number of queries, signifying the strongest relationship between the two. This demonstrates that employing $ts$ as a metric for measuring the number of vector queries yields optimal effectiveness.

Additionally, to verify the stability of the transparency parameter ts under different HNSW configurations, we calculated the Pearson correlation coefficient between the ts value of each vector and its actual query frequency for different parameter combinations. The results are shown in Table 3.

Table 3: Correlation coefficients ($\rho$) between the query count and $ts$ under different values of $M$ and efConstruct.

| $M$ | efConstruct = 50 | efConstruct = 100 | efConstruct = 150 |
|---|---|---|---|
| $M = 4$ | -0.6969 | -0.6886 | -0.6885 |
| $M = 8$ | -0.6978 | -0.6732 | -0.6603 |
| $M = 12$ | -0.6911 | -0.6696 | -0.6572 |
| $M = 16$ | -0.6860 | -0.6731 | -0.6600 |

These results indicate that, under all parameter settings, $ts$ maintains a significant negative correlation with vector query frequency (Pearson coefficients all below -0.66), suggesting that $ts$ is a stable and generalizable transparency metric that can reliably guide carrier vector selection.

In this section, we confirm the existence of transparent vectors in HNSW and draw the following conclusions through analysis and experiments:

- Some vectors that are far away from other vectors may have fewer edges or even no edges due to the edge pruning strategy, which ultimately makes them difficult to be searched.

- The transparency of a vector can be measured by its transparency score $ts$. The larger $ts$ is, the more likely it is a highly transparent vector.

# D   Comparison experiment

To conduct a more comprehensive analysis of the impact of TVP on query behavior, we conducted experiments and recorded the variance and maximum values of missed queries (MQ) and false queries (FQ) under each scheme. The results are shown in Table 4.

Table 4: Embedded impacts of different schemes.

| Scheme | SCPW(var/max)[29] | TabularMark[36] | RS | TVP |
|---|---|---|---|---|
| AMQ | 161.79 / 91.3 | 179.18 / 87.3 | 184.36 / 89.2 | **13.75 / 28.08** |
| AFQ | 187.54 / 100.9 | 184.87 / 101.6 | 189.57 / 104.7 | **12.39 / 30.15** |

As can be seen from Table 4, the fluctuations in MQ and FQ caused by TVP are significantly smaller than those caused by other methods, with both variance and maximum values being notably lower. This indicates that TVP more effectively controls query errors, not only achieving a smaller average value but also minimizing the impact in worst-case scenarios, thereby demonstrating greater stability.

Since "transparent" vectors (i.e., vectors with relatively low query frequencies) are common in any dataset, we speculate that the TVP method has good cross-dataset generalization capabilities.

To validate this hypothesis, we conducted extended experiments on two widely used deep learning embedding datasets: arxiv-nomic-768-normalized (derived from VIBE, representing text summaries) and deep1b (a classic large-scale image vector dataset). The experimental parameters were consistent with those in the paper. We evaluated the average missed queries (AMQ) and average false queries (AFQ) caused by different watermark embedding schemes, with experimental results shown in Tables 5 and 6.

Table 5: Embedded impacts of different schemes (Deep1b).

| Scheme | SCPW[29] | TabularMark[36] | RS | TVP |
|---|---|---|---|---|
| AMQ | 13.96 | 13.38 | 13.42 | **3.17** |
| AFQ | 14.21 | 13.73 | 13.59 | **3.29** |

Table 6: Embedded impacts of different schemes (arxiv-nomic-768-normalized).

| Scheme | SCPW[29] | TabularMark[36] | RS | TVP |
|---|---|---|---|---|
| AMQ | 13.51 | 13.83 | 13.75 | **4.26** |
| AFQ | 14.17 | 14.02 | 14.11 | **4.74** |

It can be observed that, on both datasets, the TVP scheme achieves significantly lower average missed query counts (AMQ) and average false positive query counts (AFQ) compared to other methods. This indicates that the strategy of prioritizing "transparent" vectors for modification can also reduce query errors on other datasets, thereby validating the cross-dataset generalization capability of our method.

# E    Method Validation And Parameter Analysis Experiment

## E.1    TVP Strategy Effectiveness

We evaluated the average number of missed queries (AMQ) and average number of false queries (AFQ) introduced by TVP under various HNSW parameter settings ($M \in \{4, 8, 12, 16\}$ and $efConstruct \in \{50, 100, 150\}$), with the number of retrieved neighbors fixed at $k = 100$. The results are summarized in Table 7.

Table 7: Embedding effects of TVP under different parameter configurations

| efConstruct | 50 (AMQ / AFQ) | 100 (AMQ / AFQ) | 150 (AMQ / AFQ) |
|---|---|---|---|
| **M = 4** | 4.69 / 4.84 | 4.32 / 4.50 | 3.71 / 3.85 |
| **M = 8** | 3.33 / 3.20 | 3.19 / 3.22 | 3.02 / 3.15 |
| **M = 12** | 2.42 / 2.45 | 2.64 / 2.61 | 2.81 / 2.83 |
| **M = 16** | 2.88 / 2.83 | 2.61 / 2.66 | 2.58 / 2.61 |

It can be seen that regardless of how the HNSW parameters change, the AMQ/AFQ values of TVP remain at a low level, indicating minimal impact on queries. This demonstrates that TVP has good adaptability to different parameter configurations.

We conducted a supplementary evaluation of TVP's impact on standard approximate nearest neighbor (ANN) retrieval performance, focusing on recall metrics for vector databases. Specifically, we measured recall at $k$ ($R@k$), which represents the proportion of true nearest neighbors successfully retrieved within the top $k$ results. A higher $R@k$ indicates greater retrieval accuracy. We compared the recall of the original, unembedded watermark database with that of the database after embedding watermarks using the TVP method, summarizing the results in Table 8.

Table 8: Changes in ANN recall rates before and after TVP watermark embedding

| R@k | 1 | 10 | 30 | 50 | 100 |
|---|---|---|---|---|---|
| Original | 0.9933 | 0.9647 | 0.8918 | 0.8253 | 0.6847 |
| TVP | 0.9914 | 0.9633 | 0.8898 | 0.8219 | 0.6843 |

As shown in the table, recall remains nearly unchanged at all $k$ levels after watermark embedding. For instance, at $R@1$, recall decreased only from 0.9933 to 0.9914, a drop of less than 0.2%. Similarly, at $R@100$, the change was negligible, from 0.6847 to 0.6843. These results indicate that TVP has a minimal impact on retrieval quality.

Table 9: Comparison of TVP and RS across different ANN index types

| Index Type | TVP (AMQ / AFQ) | RS (AMQ / AFQ) |
|---|---|---|
| NSG | 0.12 / 0.09 | 0.38 / 0.27 |
| LSH | 0.75 / 0.95 | 2.27 / 2.49 |
| IVFFlat | 1.46 / 1.52 | 3.57 / 3.81 |
| IVFPQ | 6.46 / 9.34 | 17.38 / 19.21 |

We further validated the generalization capability of the Transparent Vector Priority (TVP) strategy by applying it to multiple Approximate Nearest Neighbor (ANN) index structures. We conducted experiments on four typical indexes: NSG, LSH, IVFFlat, and IVFPQ. By recording the query frequency of each vector, we prioritized embedding watermarks in low-frequency vectors (TVP) and compared this approach with a baseline method that randomly selects vectors for embedding (RS). The experiments evaluated the performance of both schemes on average missed queries (AMQ) and average false queries (AFQ), with results shown in Table 9.

As shown in the table, TVP achieves lower AMQ and AFQ than RS across all index structures. This indicates that TVP is not only applicable to HNSW but also demonstrates excellent adaptability across index structures.

## E.2   Embedding impact

The most critical parameter affecting the embedding effect is the threshold $\tau$, which determines the transparency requirement of the algorithm for the watermark carrier vectors. A smaller value of $\tau$ implies a higher transparency requirement and fewer queries for the selected carrier vectors, which reduces the number of false queries and missed queries due to watermark embedding.

Under the premise of keeping other parameters unchanged, we adjust the parameter $th$ to study its effect on the watermark embedding effect. For each set value of $th$, the watermark embedding algorithm is executed, and the changes in the near-neighbor response domains of the carrier vectors before and after embedding are recorded, the number of false and missed queries are counted, and their averages are finally calculated.

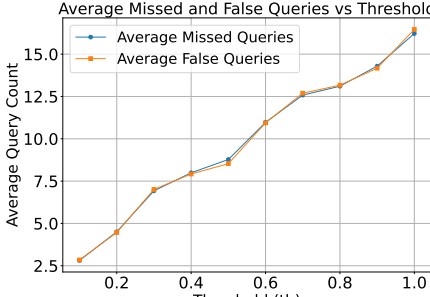

Figure 7: The influence of $\tau$ on embedding impact.

The experimental results are shown in Figure 7, where the horizontal axis represents the threshold $\tau$ and the vertical axis represents the number of queries, and the two curves correspond to the average number of false queries (AFQ) and the average number of missed queries (AMQ), respectively. It can be seen that when $\tau$ is small, the values of AFQ and AMQ are also small, which indicates that the impact of watermark embedding on the query results can indeed be kept at a very low level when the transparency requirement is high. As $th$ increases, AFQ and AMQ also increase synchronously, indicating that the $th$ parameter can effectively regulate the impact of watermark embedding, which is a key regulating factor. In addition, the figure presents an interesting phenomenon: the values of AFQ and AMQ are close to equal. We hypothesize that this is because the number of times the carrier vectors are queried does not change significantly before and after the watermark embedding, and thus the number of false queries and missed queries are roughly equal. Through experiments, we verify this speculation and explain the phenomenon in the figure.

The parameter $s$ indicates the proportion of vectors in each group that can be correctly mapped to the corresponding watermark bit after embedding the watermark. Before embedding the watermark, the mapping results are usually more uniform across the groups, with equal proportions of 1 and 0 vectors. Therefore, if $s$ is large, it is necessary to increase the proportion of vectors in a certain category from 0.5 to a very high one, which will result in more vectors needing to be modified, thus having a large impact on the query results.

## E.3   Watermark robustness

This section explores the effect of algorithm parameters on the robustness of watermarking. Based on the threat model in Section 3.4, the attacker is aware of our watermarking method and thus will adopt an adaptive attack strategy of transparent vector prioritization, e.g., prioritizing the deletion of transparent vectors when deleting vectors to remove the watermark more effectively. All the attacks discussed in this section are such adaptive attacks.

**Adaptive Deletion Attack**: An adaptive modification attack is an attempt by the attacker to remove the watermark by deleting a certain percentage of transparent vectors. The larger the parameter $s$ is, the more vectors are mapped to the corresponding watermark in each group after embedding

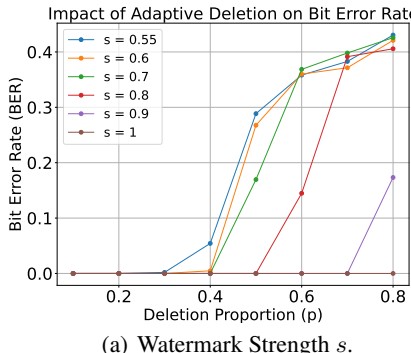
(a) Watermark Strength $s$.

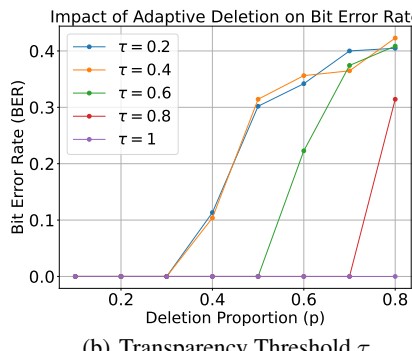
(b) Transparency Threshold $\tau$.

Figure 8: Impact of Parameters on Watermark Robustness (Under Adaptive Deletion Attacks).

the watermark, and theoretically, the more robust it should be against the adaptive deletion attack. To verify this hypothesis, we conducted experiments under the condition of $\tau = 0.5$ for different combinations of adaptive deletion proportion $p$ and watermark strengths $s$, embedded the watermark using the parameter $s$, and then deleted the vectors of the most transparent $p$ proportion, and then extracted the watermarks and computed the BER.

The experimental results are shown in Figure 8(a). The horizontal axis represents the deletion proportion $p$, the larger $p$ means the stronger the attack intensity; the vertical axis represents the BER, the larger the BER means the more extraction errors and the worse the robustness. The lines of different colors represent different $s$ values.

The following points can be observed from the Figure 8(a):

- Performance under no attack: for all curves, when the deletion proportion $p = 0$, the BER is also 0. This indicates that the watermark can be extracted completely and accurately without attack, which proves that the basic embedding and extracting functions of this scheme are reliable.

- Relation between Attack Strength and BER: as the deletion proportion $p$ increases, the BER also rises, which indicates that adaptive deletion is indeed an effective means of attack.

- Effect of watermark strength $s$ on robustness: for the same strength of deletion attack, the larger the $s$, the lower the BER. This indicates that larger values of $s$ can enhance the robustness of watermarking, which is consistent with our speculation.

- High robustness: when $s$=1, the BER always remains 0, which means that the watermark can be extracted accurately no matter how strong the attack is. This result shows that the proposed scheme is very robust at $s$=1 and can meet the needs of application scenarios that require high robustness.

Next, we discuss the impact of the transparency threshold $\tau$ on the robustness. $\tau$ determines the requirement of the algorithm on the transparency of the carrier vectors. The smaller $\tau$ is, the higher the transparency of the selected vectors, and the more obvious their features are, which are easier to be recognized and suffer from accurate attacks under adaptive attacks. Therefore, the smaller $\tau$ is, the less robust the watermark should be.

To verify this hypothesis, we conducted experiments under the condition of $s = 0.7$ for different combinations of adaptive deletion proportion $p$ and thresholds $th$, embedded the watermark using the parameter $\tau$, and then deleted the vectors of the most transparent $p$ proportion, then extracted the watermark and computed the BER.

The experimental results are shown in Figure 8(b). The horizontal axis represents the deletion proportion $p$, which indicates the strength of the attack; the vertical axis represents the BER. The lines of different colors represent different $\tau$ values. It can be seen that for the same deletion proportion, the smaller $\tau$ is, the larger the BER is, which is consistent with our hypothesis. In addition, the curves of $\tau = 0.2$ and $\tau = 0.4$ almost overlap in the figure. This indicates that the setting of the complementary vectors plays a role in the selection of the carrier vectors. When $\tau$ is too small and $s$

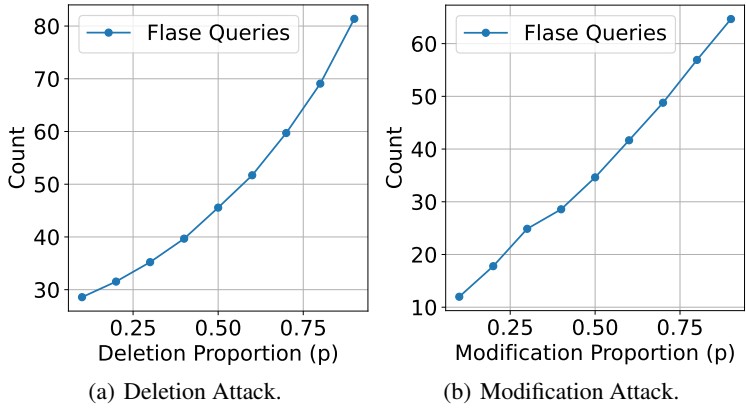

(a) Deletion Attack.    (b) Modification Attack.

Figure 9: Impact of Attacks on Query Results.

is large, the number of vectors satisfying the transparency requirement may be insufficient. In this case, the algorithm, after prioritizing the high transparency vectors, will supplement the selection of some vectors with lower transparency. Therefore, although $\tau$ is different, the selected vectors are similar in this case, and the $BER$ are similar when facing adaptive deletion attacks.

According to the analysis of the threat model, an attacker needs to balance the effect of watermark removal with its impact on their own use of the database. To this end, we test the specific impact of adaptive deletion attacks on query results. The experiment sets the number of query neighbors $k = 100$ and measures the average number of query errors of all vectors before and after deleting varying proportions of vectors, and the experimental results are shown in Figure 9(a). The figure shows that the number of query errors rises significantly as the deletion proportion increases. When the deletion rate reaches 0.5, the average number of query errors is close to 50, which means that the proportion of query errors is close to half of the total number of queries. At this intensity, the query result has been seriously distorted, and it is difficult to meet the actual application requirements. Therefore, attackers usually do not choose such a high intensity attack, and it can be inferred that the actual attack intensity is lower than the maximum value set by the experiment.

**Adaptive Modification Attack**: An adaptive modification attack is an attempt by the attacker to remove the watermark by modifying a certain percentage of transparent vectors, where the modifications are conducted on the partial dimension values of the vectors.

Under the condition of threshold $\tau = 0.5$, we conducted experiments for different combinations of watermark strength $s$ and modification proportion $p$. The specific steps include watermark embedding, performing modification attacks, and then extracting the watermark and recording the $BER$. The test results show that a significant bit error rate occurs only when the modification dimension reaches 30. Therefore, we set the number of modified dimensions for the modification attack to 30.

The experimental results are shown in Figure 10(a). The horizontal axis represents the modification attack strength, and the vertical axis represents the BER. The lines of different colors represent different $s$ values. It can be seen that for the same modification proportion, the larger the $s$, the smaller the BER, indicating the higher robustness. This verifies that a higher value of $s$ can significantly enhance the modification resistance of the watermark.

Next, we test the effect of the transparency threshold $\tau$ on watermark robustness. Under the condition of watermark strength $s = 0.7$, we conducted experiments for different combinations of transparency threshold $th$ and modification proportion $p$. The experimental steps also include watermark embedding, performing a modification attack, and then extracting the watermark followed by recording the $BER$.

The experimental results are shown in Figure 10(b). The horizontal axis represents the strength of the modification attack (i.e., modification proportion $p$) and the vertical axis represents the $BER$. The lines of different colors represent different $\tau$ values. It can be seen that with the same modification proportion, the larger the $\tau$, the smaller the BER, indicating the higher robustness.

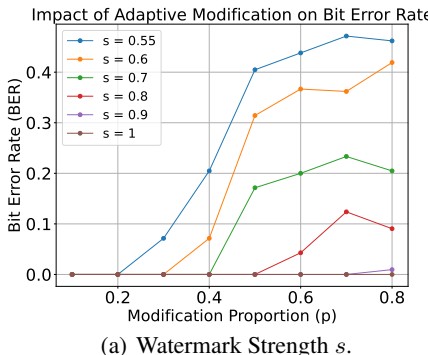
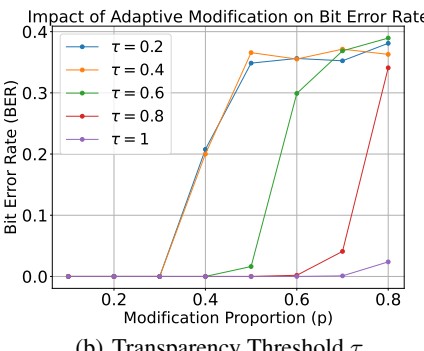

(a) Watermark Strength $s$.

(b) Transparency Threshold $\tau$.

Figure 10: Impact of Parameters on Watermark Robustness (Under Adaptive Modification Attacks).

Finally, we evaluated the impact of this attack on query results. We measured the average number of query errors for all vectors before and after modifying different proportions of vectors. The results are shown in Figure 9(b). As the proportion of modified vectors increases, the number of false queries increases rapidly. On the other hand, since the watermarking algorithm modifies only one dimension for each vector, and the attack needs to modify multiple dimensions to significantly impact watermark extraction, it indicates that the modification attack is more expensive to implement. Therefore, to ensure database availability, attackers usually avoid high-intensity adaptive modification attack strategies, which limits the actual threat of the attack.

**Shuffle Attack**: Shuffle Attack is to disturb the order between vectors and the correspondence between vectors and IDs. The watermark extraction of TVP does not depend on the order of vectors or their original IDs, so this attack is not effective for TVP.

**Reconstructed Index Attack**: Reconstruction indexing attack refers to reconstructing the indexes of vectors. Since the index construction process has a certain degree of randomness, even if the same vectors are used to construct the index multiple times, the results will be different. If the watermark is embedded in the index structure, reconstructing the index will lead to wrong watermark extraction. The TVP scheme does not depend on the index structure, so the reconstructed index attack is also ineffective for TVP.

By analyzing the experiments in Section E.2 and Section E.3, we draw the following brief conclusions:

- **Transparency Threshold** $\tau$: the smaller $\tau$ is, the smaller the impact on query result is, but the robustness will become worse.
- **Watermark Strength** $s$: the larger $s$ is, the stronger the robustness is, but more vectors need to be modified and the impact on query results increases.

