# OpenReview forum: "Vector Database Watermarking"
_NeurIPS.cc/2025/Conference — NeurIPS 2025 poster_

### Official Review · Reviewer_SNQH · 2025-06-09

**Clarity:** 3
**Significance:** 3
**Originality:** 3
**Rating:** 4
**Confidence:** 3

**Summary:**

The paper introduces the innovative Transparent Vector Priority (TVP) watermarking scheme for vector databases, effectively protecting these assets from unauthorized replication while preserving ANN query accuracy. By leveraging the HNSW structure’s edge properties, TVP prioritizes low-query-frequency transparent vectors, reducing query errors by ~75% compared to existing methods, as demonstrated on the ANN_SIFT1M dataset. Its robust design against various attacks and commitment to open access for code and data highlight a significant contribution to secure vector database management.

**Questions:**

1. The TVP scheme uses basic graph metrics like degree and edge length. Have the authors considered more refined measures, such as spectral properties or centrality, to improve carrier selection and theoretical grounding?
2. The experiments are conducted with fixed HNSW parameters (\(M=8\), \(efConstruct=100\)). How does the TVP scheme perform under different HNSW configurations (e.g., varying \(M\) or \(efConstruct\)), and can the authors validate the robustness of the transparency parameter (ts) across these variations?
3. Given the reliance on ANN_SIFT1M, have the authors tested or considered evaluating TVP on other datasets, such as deep1b dataset, to ensure the generalizability of the transparent vector selection strategy?

**Ethical Concerns:**

["NO or VERY MINOR ethics concerns only"]

**Final Justification:**

In light of the discussions, I consider my concerns adequately resolved and recommend a borderline accept.

**Limitations:**

yes

**Quality:**

3

**Strengths And Weaknesses:**

Strength:
The paper’s Transparent Vector Priority (TVP) scheme innovatively uses HNSW’s graph properties to embed watermarks in low-query-frequency vectors, reducing ANN query errors by ~75% and ensuring robustness against attacks, with reproducible results on ANN_SIFT1M.

Weakness:
1. Limited Graph-Theoretic Depth: TVP relies on basic node degree and edge length metrics for transparent vector selection, lacking advanced graph-theoretic analysis (e.g., spectral or centrality measures), reducing its innovation in graph-based watermarking.
2. Narrow Experimental Scope: Experiments use fixed HNSW parameters (\(M=8\), \(efConstruct=100\)) and a single dataset (ANN_SIFT1M), limiting validation of TVP’s applicability across varied configurations and datasets.

---

> ### Author Rebuttal · Authors · 2025-07-29
>
> Dear Reviewer SNQH:
>
> We sincerely appreciate your constructive comments and insightful observations. In response to your suggestions, we have supplemented our experiments and further clarified the theoretical details. If our response has addressed your concerns, we kindly ask that you consider raising your rating, for which we would be most grateful. If you have any further suggestions or concerns, we will continue to respond positively and improve our work.
>
> ## **Weakness 1 & Question 1.**
> > **The TVP scheme uses basic graph metrics like degree and edge length. Have the authors considered more refined measures, such as spectral properties or centrality, to improve carrier selection and theoretical grounding?**
>
> Thank you for your valuable suggestions.
> Currently, the TVP strategy primarily selects carrier vectors based on the fundamental structural characteristics of the HNSW graph, utilizing each node's degree and edge length to construct transparency scores. This method is efficient in implementation and has already demonstrated significant performance improvements. In fact, node degree has been widely regarded as a measure of centrality in graph theory. TVP has preliminarily embodied the concept of degree centrality and further enhanced the discrimination of “transparency” through edge length information.
> We fully agree with the reviewers' perspective that more refined graph theory metrics can uncover nodes' deeper structural roles within the graph, potentially providing a more robust theoretical foundation and selection precision for carrier selection.
> In future work, we plan to systematically incorporate these more complex graph metrics to further enhance TVP's theoretical rigor and application performance.
>
> ## **Weakness 2 & Question 2.**
>
> > **The TVP scheme's performance and the robustness of the transparency parameter (ts) under different HNSW configurations.**
>
> Thank you for your interest in the stability and generalization capabilities of TVP. We fully agree on the necessity of verifying the robustness of the method under different HNSW configurations.
>
> Based on your feedback, we evaluated the average number of missed queries (AMQ) and average number of false queries (AFQ) introduced by TVP under various HNSW parameter settings (\$M \in \\{4, 8, 12, 16\\}\$ and \$efConstruct \in \\{50, 100, 150\\}\$), with the number of retrieved neighbors fixed at \$k = 100\$. The results are summarized in Table 1.
>
> Table 1: Embedding effects of TVP under different parameter configurations
> | efConstruct | 50 (AMQ / AFQ) | 100 (AMQ / AFQ) | 150 (AMQ / AFQ) |
>  |----|-------------------|--------------------|--------------------|
> |M = 4 | 4.69 / 4.84 | 4.32 / 4.50 | 3.71 / 3.85 |
> |M = 8 | 3.33 / 3.20 | 3.19 / 3.22 | 3.02 / 3.15 |
> |M = 12 | 2.42 / 2.45 | 2.64 / 2.61 | 2.81 / 2.83 |
> |M = 16 | 2.88 / 2.83 | 2.61 / 2.66 | 2.58 / 2.61 |
>
> It can be seen that regardless of how the HNSW parameters change, the AMQ/AFQ values of TVP remain at a low level, indicating minimal impact on queries. This demonstrates that TVP has good adaptability to different parameter configurations.
> Additionally, to verify the stability of the transparency parameter ts under different HNSW configurations, we calculated the Pearson correlation coefficient between the ts value of each vector and its actual query frequency for different parameter combinations. The results are shown in Table 2.
>
> Table 2: Pearson correlation coefficient between the ts of vectors and the number of times they are queried under different parameter configurations.
>
> | efConstruct | 50 | 100 | 150 |
> |----|------------------|-------------------|-------------------|
> | M = 4 | -0.6969 | -0.6886 | -0.6885 |
> | M = 8 | -0.6978 | -0.6732 | -0.6603 |
> | M = 12 | -0.6911 | -0.6696 | -0.6572 |
> | M = 16 | -0.6860 | -0.6731 | -0.6600 |
>
> These results indicate that, under all parameter settings, ts maintains a significant negative correlation with vector query frequency (Pearson coefficients all below -0.66), suggesting that ts is a stable and generalizable transparency metric that can reliably guide carrier vector selection.
>
> In summary, we believe that TVP continues to demonstrate consistent advantages under different HNSW parameter configurations, and its core idea (screening transparent vectors via ts) exhibits excellent robustness and generalization capabilities.
>
> ## **Question 3.**
>
> > **Experiments on other datasets.**
>
> Since “transparent” vectors (i.e., vectors with relatively low query frequencies) are common in any dataset, we speculate that the TVP method has good cross-dataset generalization capabilities.
>
> To validate this hypothesis, we conducted extended experiments on two widely used deep learning embedding datasets: arxiv-nomic-768-normalized (derived from VIBE, representing text summaries) and deep1b (a classic large-scale image vector dataset). The experimental parameters were consistent with those in the paper. We evaluated the average missed queries (**AMQ**) and average false queries (**AFQ**) caused by different watermark embedding schemes, with experimental results shown in Tables 3 and 4.
>
> Table 3: Experimental results on the Deep1b dataset
>
> | Scheme | SCPW | TabularMark | RS | TVP |
>  |------------|------|-------------|-----|------|
>  | **AMQ** | 13.96| 13.38 | 13.42| **3.17** |
> | **AFQ** | 14.21| 13.73 | 13.59| **3.29** |
>
> Table 4: Experimental results on the arxiv-nomic-768-normalized dataset
> | Scheme | SCPW | TabularMark | RS | TVP |
> |------------|------|-------------|-----|------|
> | **AMQ** | 13.51| 13.83 | 13.75| **4.26** |
> | **AFQ** | 14.17| 14.02 | 14.11| **4.74** |
>
> It can be observed that, on both datasets, the TVP scheme achieves significantly lower average missed query counts (AMQ) and average false positive query counts (AFQ) compared to other methods. This indicates that the strategy of prioritizing “transparent” vectors for modification can also reduce query errors on other datasets, thereby validating the cross-dataset generalization capability of our method.

---

> > ### Comment · Reviewer_SNQH · 2025-08-04
> >
> > I thank the authors for their detailed rebuttal effort. Based on the positive experiment result on transparency parameters and some results on new datasets, I am willing to keep my rating unchanged.

---

> > > ### Author Response · Authors · 2025-08-04
> > >
> > > We sincerely thank you for carefully reading our rebuttal and acknowledging our experimental results. The experiments and discussions involved in the rebuttal phase will be included in the final revised version. Thank you again for your careful review.

---

### Official Review · Reviewer_ieq6 · 2025-06-26

**Clarity:** 2
**Significance:** 2
**Originality:** 2
**Rating:** 4
**Confidence:** 3

**Summary:**

The proposed method deals with embedding watermark information into feature vectors to prevent unauthorized replication of the vector DB.
Each data point belongs to one of L groups and is converted to a binary value by cryptographic mapping.
Then the ratio of 1-vectors to 0-vectors in the g-th group determines the g-th value of the watermark code.
If the watermark code is pre-defined, a portion of bits in $n_m$ vectors must be flipped to match the ratio of 0- or 1-vectors to the watermark code.
The main contribution of the proposed method is that it selects the data to be flipped based on their expected lower search frequency, rather than choosing them randomly.

**Questions:**

[Question 1]
As mentioned in Weakness 1, this method inevitably introduces performance degradation.
However, since the paper does not use standard ANN evaluation metrics such as R@1, it is difficult to quantify the extent of this degradation.
The authors should report these metrics.

[Question 2]
As mentioned in Weakness 2, it would be beneficial to provide the variance and maximum values of MQ and FQ instead of the average.
Additionally, the paper should show the difference between queries whose first nearest neighbor is a modified data point and those whose first nearest neighbor is an unmodified data point.

[Question 3]
As mentioned in Weakness 3, bit flips can completely alter the feature information when used together with VQ techniques like product quantization.
Since HNSW and VQ techniques are often used in combination, an ablation study on this case should be provided.

[Question 4]
If the watermark code S is chosen according to the original cryptographic value distribution, wouldn’t it be possible to skip the modification step?

**Ethical Concerns:**

["NO or VERY MINOR ethics concerns only"]

**Final Justification:**

The authors provided additional experiments that addressed the concerns for Q1 and Q2.
A detailed explanation for Q4 was helpful for understanding, and although Q3 remains unresolved, it is not critical enough to warrant rejection.
Therefore, I am raising my score from 3 to 4.

**Limitations:**

yes

**Paper Formatting Concerns:**

No problems were observed.

**Quality:**

2

**Strengths And Weaknesses:**

Strengths

1. The proposed method appears to be robust against deletion attacks because altering a specific g-th watermark bit requires changing the majority in the g-th group, which cannot be achieved by deleting only a few data points.

2. The paper provides a clear introduction to the important yet underexplored area of preventing unauthorized replication in vector databases.

3. Although the paper states that the method is limited to HNSW, it appears to be applicable to other indexing structures as long as the method for determining the transparency parameter is modified.

Weaknesses

1. However, the method inherently modifies the original feature vectors, which inevitably leads to performance degradation.
It would be more ideal to embed watermark information without altering the original feature vectors, such as by using the connectivity of the ANN graph or the indices of nodes selected in the upper layers of HNSW.

2. The modifications are applied to vectors that are expected to be searched less frequently, so the overall performance degradation is minimal.
However, if a query with the modified vector as its first nearest neighbor is provided as input, significant performance degradation can occur.

3. The impact of bit flips during the modification stage is highly concerning when used together with VQ.
While bit flips have little effect on data measured by Hamming distance, they have a much greater impact on data measured by L2 distance.
Moreover, when each bit of the feature refers to a pre-defined codebook index, as in VQ, this impact is further amplified.

---

> ### Author Rebuttal · Authors · 2025-07-29
>
> Dear Reviewer ieq6:
>
> We sincerely appreciate your insightful and constructive feedback. In response to your comments, we conducted a series of additional experiments and analyses, including evaluating standard ANN recall metrics (e.g., R@1), the variance and maximum value of query errors, and the compatibility of our method with vector quantization (VQ) technology.
> These efforts reflect our deep respect for your feedback. If our modifications meet your expectations, we would be grateful if you could consider raising your rating. If there are any further questions or concerns, we welcome your continued suggestions, and we will actively work to further improve the paper.
>
> ## **W1 & Q1.**
> > **Modifying vectors will result in performance degradation. Standard ANN evaluation metrics such as R@1 should be added.**
>
> We have conducted a supplementary assessment of the impact of TVP on standard ANN metrics, recording the recall rates of the vector database without watermarks and after embedding watermarks. The experimental results are shown in Table 1.
>
> Table 1: Changes in ANN recall rates before and after TVP watermark embedding
> | R@k | 1 | 10 | 30 | 50 | 100 |
> |-------|-------|-------|-------|-------|-------|
>  | Original | 0.9933 | 0.9647 | 0.8918 | 0.8253 | 0.6847 |
> | TVP | 0.9914 | 0.9633 | 0.8898 | 0.8219 | 0.6843 |
>
> It can be seen that the impact of TVP on all recall metrics is very limited, with minimal performance degradation.
> We completely understand your concern about “performance loss due to embedding.” This is a common issue faced by most perturbation-based watermarking schemes, not unique to TVP. However, unlike the “using the connectivity of the ANN graph” method, perturbation-based watermarking offers greater robustness, making it difficult for attackers to remove the watermark through simple index reconstruction. This provides a more reasonable trade-off between security and practicality.
>
> ## **W2 & Q2.**
> > **As mentioned in Weakness 2, it would be beneficial to provide the variance and maximum values of MQ and FQ instead of the average.**
>
> Thank you for your valuable suggestions. To conduct a more comprehensive analysis of the impact of TVP on query behavior, we conducted experiments and recorded the variance and maximum values of missed queries (MQ) and false queries (FQ) under each scheme. The results are shown in Table 2.
>
> Table 2: Variance/Maximum Values of MQ and FQ for Different Schemes
>
> |  | SCPW (Var / Max) | TabularMark    | RS             | TVP               |
> | -------- | ---------------- | -------------- | -------------- | ----------------- |
> | **MQ**   | 161.79 / 91.3    | 179.18 / 87.3  | 184.36 / 89.2  | **13.75 / 28.08** |
> | **FQ**   | 187.54 / 100.9   | 184.87 / 101.6 | 189.57 / 104.7 | **12.39 / 30.15** |
>
> As can be seen from the table, the fluctuations in MQ and FQ caused by TVP are significantly smaller than those caused by other methods, with both variance and maximum values being notably lower. This indicates that TVP more effectively controls query errors, not only achieving a smaller average value but also minimizing the impact in worst-case scenarios, thereby demonstrating greater stability.
>
> ## **W2 & Q2 Follow-up.**
> > **Although the overall impact is small, performance may degrade significantly when the modified vector becomes the top-1 nearest neighbor. Comparison results between such queries and regular queries should be provided.**
>
> Thank you for your insightful comments. We fully agree that analyzing specific queries where the Top-1 nearest neighbor is the modified vector can provide a deeper understanding of potential performance degradation.
>
> To address this issue, we conducted an additional experiment, dividing all queries into two groups:
>
> * **M group (modified)**: Queries where the Top-1 nearest neighbor is the modified vector;
> * **U group (unmodified)**: Queries where the Top-1 nearest neighbor is the unmodified vector.
>
> We then compared the recall rates (R\@k) of the two groups. The results are shown in table 3.
>
> Table 3: Nearest neighbor query performance of the two vector groups.
>
> | k   | Modified R@k | Unmodified R@k | Difference |
> | --- | ------------- | --------------- | ---------- |
> | 1   | 0.9545        | 0.9998         | 0.0453     |
> | 10  | 0.9682        | 0.9692          | 0.0010     |
> | 30  | 0.9000        | 0.8940          | -0.0060    |
> | 50  | 0.8336        | 0.8251          | -0.0085    |
> | 100 | 0.6914        | 0.6847          | -0.0066    |
>
>
> Although Group M shows a slight decrease in performance at R@1 (approximately 4.5%), its performance rapidly converges with Group U as k increases. It is worth noting that only 4.61% of queries belong to Group M, indicating that after adopting the “transparent vector priority” strategy, very few modified vectors become the first nearest neighbor. Therefore, even though there is a certain performance difference at R@1, its impact on the overall retrieval effectiveness remains limited.
>
> ## **W3 & Q3.**
> > **As mentioned in Weakness 3, bit flips can completely alter the feature information when used together with VQ techniques like product quantization. Since HNSW and VQ techniques are often used in combination, an ablation study on this case should be provided.**
>
> Thank you for pointing out the potential serious implications of bit flipping in a vector quantization (VQ) environment. This is a critical issue that has prompted us to further explore the compatibility between watermark embedding and VQ technology.
>
> We understand your concerns as follows: In vector databases using VQ techniques such as product quantization (PQ), each subvector is typically encoded as a codeword (i.e., an index in the codebook), which corresponds to a specific centroid vector. If we directly flip the bits of these discrete indices, although the modification seems small from the perspective of Hamming distance, the large Euclidean distance between the corresponding centroids of adjacent indices will lead to significant L2 distance deviation, which may seriously affect the ANN query results. Therefore, simple bit flipping is no longer applicable in this scenario.
>
> We agree with this view and recognize that in vector databases combining VQ technology, watermark embedding should avoid direct bit-level perturbations. We have an initial idea for this:
> When embedding watermarks, we can treat each dimension's codebook as a candidate set, first read the centroid vector corresponding to the current codeword, and then sort the other candidate centroid vectors in ascending order of their L2 distances from the current codeword. Among these candidates with closer distances, we can select an alternative codeword that both maps to the target watermark bit and is as close as possible to the original center vector, thereby minimizing the impact of modifications. This is a “minimum perturbation” watermark embedding strategy specifically tailored for scenarios combining VQ technology.
>
> Additionally, we believe that since bit flipping is not applicable to VQ-encoded vectors, conducting ablation experiments in this environment is not meaningful and may lead to misunderstandings. We will explicitly highlight the limitations of the bit flipping strategy in the discussion section of the paper and will explore watermark design tailored to VQ scenarios as an important direction for future research.
>
> Once again, thank you for raising this forward-thinking and insightful question.
>
> ## **Q4.**
> > **If the watermark code S is chosen according to the original cryptographic value distribution, wouldn’t it be possible to skip the modification step?**
>
> Thank you for your insightful question. We understand that you mean: if the pre-selected watermark code $S$ happens to match the distribution of values naturally extracted from the vector database, then perhaps we can skip the modification step and directly use $S$ as the “natural watermark” of the original database.
>
> This approach does indeed have potential in certain application scenarios, particularly in data integrity verification. For example, data owners could record $S$ after its initial extraction and use it in the future to verify whether the database has been modified. However, in the copyright authentication scenario we are focusing on, this approach has limitations: the naturally extracted $S$ is typically a string of meaningless random bits lacking semantic or structural meaning, and thus cannot fulfill the functions of identifying ownership or conveying copyright information. Therefore, the modification process remains a necessary step for embedding specified watermark information.
>
> Nevertheless, your question has inspired us to explore a potential optimization direction: introducing additional parameters to guide the original distribution toward the target watermark, thereby reducing the number of vectors requiring modification. We will further investigate this approach in future work.

---

> > ### Comment · Reviewer_ieq6 · 2025-08-04
> >
> > Thank you to the authors for providing additional experiments and a detailed response.
> > The additional experiments have successfully addressed my concerns regarding Q1 and Q2.
> > I also appreciate the detailed explanation for Q4, which clarified the underlying points and improved my understanding of the authors’ approach.
> > (If I understood correctly, the S must be an artificial-looking string in order to claim copyright.)
> > While Q3 remains unresolved, it is not critical enough to warrant rejection.
> > Based on these improvements, I have raised my score from 3 to 4.
> > I recommend incorporating the additional experiments from the rebuttal into the final version of the paper.

---

> > > ### Author Response · Authors · 2025-08-04
> > >
> > > Thank you very much for your recognition of our work and the positive evaluation. We are delighted to have addressed your concerns, and we will incorporate the additional experiments conducted during the rebuttal phase into the final version to further enhance the completeness of the paper. Once again, we sincerely appreciate your valuable suggestions and support!

---

### Official Review · Reviewer_ir5J · 2025-06-30

**Clarity:** 2
**Significance:** 2
**Originality:** 3
**Rating:** 4
**Confidence:** 3

**Summary:**

This paper introduces a watermarking scheme for vector databases, focusing on addressing the security threat of unauthorized replication. The core of the scheme is the Transparent Vector Priority (TVP) strategy, which leverages the characteristic of the Hierarchical Navigable Small World (HNSW) indexing structure where some vectors have few or no edges and thus low query frequencies. By prioritizing watermark embedding in these "transparent" vectors, TVP minimizes the impact on Approximate Nearest Neighbor (ANN) query results. Experiments show that TVP reduces missed and false queries by about 75% compared to existing schemes. The paper's contributions include proposing the first vector database watermarking scheme, defining metrics to quantify embedding impact, and optimizing the scheme via transparent vector prioritization to balance query performance and watermark robustness.

**Questions:**

1. May I ask if the phrase “Presenting RS” summarized in the Contributions section is a typo? The main technical contribution of this paper should be the TVP strategy.
2. Could the authors provide comparisons with more methods or evaluate the model's performance on additional datasets?

**Ethical Concerns:**

["NO or VERY MINOR ethics concerns only"]

**Final Justification:**

The authors' response addressed my concerns. After carefully considering both the rebuttal and the comments from other reviewers, I have decided to raise my score to 4.

**Limitations:**

1. The authors should improve the writing and structure of the paper to make it more readable and easier to understand.
2. See in Weakness.

**Paper Formatting Concerns:**

No paper formatting issues.

**Quality:**

2

**Strengths And Weaknesses:**

Strengths:
1. Proposing the first kind vector database watermarking scheme (RS) fills a gap in the research field.
2. Defining missed queries and false queries to measure the impact of watermark embedding on ANN queries is a precise and necessary step.
3. The TVP scheme shows a smart optimization approach. By leveraging the characteristics of the HNSW index structure (identifying vectors with low query frequency as "transparent vectors") to minimize the impact of watermark embedding, it addresses a key concern in watermarking—preserving the normal operation of the database.
Weakness:
1. The allocation of space in the paper is problematic: the sections Basic Methodology: Random Selection and Preliminaries are somewhat lengthy, which results in some important ablation studies being relegated to the appendix. There is still some content overlap between the main text and the appendix, such as Table 1 and Table3
2. The experimental validation is insufficient, as the experiments were conducted on only a single dataset. Moreover, the two baseline methods used for comparison were not specifically designed for vector datasets, which introduces a certain degree of unfairness.

---

> ### Author Rebuttal · Authors · 2025-07-29
>
> Dear Reviewer ir5J:
>
> We appreciate your insightful and valuable comments. We have carefully considered each comment and responded to them individually, and sincerely hope that our responses have adequately addressed your concerns. If so, we would be grateful if you could raise your score. If not, please let us know your further concerns, and we will continue to respond positively to your comments and improve our submission.
>
> ## **Weakness 1 & Limitation 1.**
> > **The allocation of space in the paper is problematic.**
>
> Thank you for your comment. We will rearrange the length of each section to ensure that the core content of each section is presented more clearly and concisely. In particular, for the sections on the “Random Selection (RS)” method and Preliminaries, we plan to streamline the content of Section 3.1 Approximate Nearest Neighbor (ANN) and Section 3.2 Hierarchical Navigable Small World (HNSW), moving more background information to the appendix. Additionally, we will explore the possibility of merging the RS and TVP schemes into a single section to free up more space in the main text for presenting more comprehensive ablation experiment results. Furthermore, we will ensure that there is no repetition between the main text and the appendix. Once again, thank you for your careful review.
>
> ## **Weakness 2 & Question 2.**
> > **The experimental validation is insufficient, as the experiments were conducted on only a single dataset.**
>
> We thank your valuable suggestions regarding the generalization of the experiment.
>
> Since “transparent” vectors (i.e., vectors with relatively low query frequencies) are common in any dataset, we speculate that the TVP method has good cross-dataset generalization capabilities.
>
> To validate this hypothesis, we conducted extended experiments on two widely used deep learning embedding datasets: arxiv-nomic-768-normalized (derived from VIBE, representing text summaries) and deep1b (a classic large-scale image vector dataset). The experimental parameters were consistent with those in the paper. We evaluated the average missed queries (**AMQ**) and average false queries (**AFQ**) caused by different watermark embedding schemes, with experimental results shown in Tables 1 and 2.
>
>
> Table 1: Experimental results on the Deep1b dataset
>
> | Scheme | SCPW | TabularMark | RS | TVP |
>  |------------|------|-------------|-----|------|
>  | **AMQ** | 13.96| 13.38 | 13.42| **3.17** |
> | **AFQ** | 14.21| 13.73 | 13.59| **3.29** |
>
> Table 2: Experimental results on the arxiv-nomic-768-normalized dataset
> | Scheme | SCPW | TabularMark | RS | TVP |
> |------------|------|-------------|-----|------|
> | **AMQ** | 13.51| 13.83 | 13.75| **4.26** |
> | **AFQ** | 14.17| 14.02 | 14.11| **4.74** |
>
> It can be observed that, on both datasets, the TVP scheme achieves significantly lower average missed query counts (AMQ) and average false positive query counts (AFQ) compared to other methods. This indicates that the strategy of prioritizing “transparent” vectors for modification can also reduce query errors on other datasets, thereby validating the cross-dataset generalization capability of our method.
>
>
> ## **Weakness 2.**
> > **Moreover, the two baseline methods used for comparison were not specifically designed for vector datasets, which introduces a certain degree of unfairness.**
>
> Since watermarking in vector databases is still an emerging topic, there are currently no mature baseline methods specifically designed for this scenario. Therefore, we selected high-quality methods that are representative of the database and dataset watermarking fields in recent years as comparison objects.
>
> These methods were originally designed for other data models and did not take into account the structural characteristics (e.g., index structure) and usage scenarios (e.g., ANN queries) of vector databases. Therefore, they may not achieve optimal performance in our experimental setup.
>
> However, this result itself highlights a critical issue: existing watermarking methods are difficult to directly apply to vector databases, and their performance in this scenario has obvious shortcomings. This phenomenon indirectly validates the necessity and innovation of our work, namely: there is an urgent need to design specialized watermarking methods tailored to the unique structure and application scenarios of vector databases, which is precisely the core contribution of the TVP and other schemes proposed in this paper.
>
> ## **Question 1.**
> > **May I ask if the phrase “Presenting RS” summarized in the Contributions section is a typo? The main technical contribution of this paper should be the TVP strategy.**
>
> Thank you for your careful correction. “Presenting RS” was indeed a typo. The main technical contribution of this paper should be the TVP strategy. We will correct this in the revised version.

---

> > ### Comment · Reviewer_ir5J · 2025-08-04
> > **Reply to authurs**
> >
> > Thank you for the detailed response and the additional experiments. After considering the authors’ rebuttal along with the comments from other reviewers, I have decided to raise my score from 3 to 4. I hope the authors can address noticeable yet trivial issues such as the typo raised by Q1 in the following version. Wish authors the best luck.

---

> > > ### Author Response · Authors · 2025-08-04
> > >
> > > We sincerely thank you for your comprehensive review of our work and valuable feedback. We are pleased that our response has addressed the issues you raised in your rebuttal and appreciate your positive evaluation after considering all perspectives. We have carefully noted the specific issues you raised (including the spelling errors mentioned in Question 1) and will thoroughly check for any potential spelling errors in the final version. Thank you again for your review and support.

---

### Official Review · Reviewer_Ncy6 · 2025-07-03

**Clarity:** 2
**Significance:** 3
**Originality:** 4
**Rating:** 4
**Confidence:** 4

**Summary:**

This paper introduces the problem of vector database watermarking. With the advent of vector embeddings as a prominent data modality, vector databases have become valuable digital assets. As a result, this paper is the first to extend the idea of watermarking, which has previously been well studied in the context of relational databases, to the new domain of vector DBs. In this paper, the authors propose two vector DB watermarking schemes: random selection (RS) and transparent vector priority (TVP). Random selection selects vectors randomly for perturbation and has the benefit of being agnostic to the specifics of the how the vector DB is implemented. However, RS has relatively poor "embeddings impact" as the perturbations can significantly degrade the quality of the vector DB results. In fact, Table 2 shows that RS performs worse than a method designed for tabular data and not vector DBs specifically. On the other hand, TVP focuses on watermarking for the specific case of the popular HNSW index. The authors propose a novel scheme for identifying "transparent vectors" -- those that are far away from others in the graph -- and seek to perturb those. In Table 2, the authors report that TVP outperforms other watermarking baselines for HNSW by a sizable margin. In summary, this paper is the first to introduce the problem of vector database watermarking and proposes two candidate solutions: random selection and transparent vector priority. The paper also discusses the tradeoffs between these watermarking approaches and presents metrics for assessing the quality of vector DB watermarks.

**Questions:**

Q1. Can you provide formal definitions of watermarking (and specifically a vector DB watermark)? *I would be willing to raise my evaluation score if the authors can revise their paper to present a formal definition and demonstrate how their proposed schemes fit within this definition.*

Q2. Can you provide a formal definition of the threat model and include a (possibly informal) proof of security associated with your proposed watermarking schemes? I am particularly interested in seeing a more formal treatment of the adversary's capabilities and computational power? For example, would an adversary with unlimited computational power be able to break these watermarking schemes? *I would be willing to raise my evaluation score if the authors can provide a formal treatment of the threat model and provide a proof of security about to their proposed schemes in relation to an adversary with precisely defined capabilities.*

Q3. Can you extend the watermarking evaluation to more datasets beyond SIFT 1M? I would particularly be interested in seeing results for deep learning based embedding datasets. [VIBE](https://github.com/vector-index-bench/vibe), [ANN Benchmarks](https://github.com/erikbern/ann-benchmarks) and [BigANN Benchmarks](https://github.com/harsha-simhadri/big-ann-benchmarks) might all be good places to obtain additional datasets. *I would be willing to raise my evaluation score if the authors can demonstrate that their proposed techniques generalize across a variety of datasets. In particular, I am curious to see if the logic behind node selection in TVP generalizes to additional datasets. I would also be curious to know of TabularMark is still on par or better than RS on additional datasets.*

**Ethical Concerns:**

["NO or VERY MINOR ethics concerns only"]

**Final Justification:**

Based on the authors' responses during the rebuttal and discussion period, I will raise my score to a 4 and will advocate for acceptance of this paper. I choose not to assign a higher score at this time because I think the paper would still benefit from a more comprehensive evaluation on more benchmark datasets (at least 5-6 in total), and I hope the authors can consider adding these additional results to the camera-ready version of the paper.

I congratulate the authors on their strong efforts to address reviewer concerns during the rebuttal and discussion period. I think their proposed additions to the paper (formal definitions of watermarking and the threat model, discussions around the adversarial robustness of their AMQ and AFQ metrics, and more thorough experiments on additional datasets and indexes) make the paper much, much stronger than their initial submission. Thus, I am happy to recommend acceptance of this paper. I believe the idea of vector database watermarking is a new and timely contribution to the community and has the potential to inspire additional research in this direction.

**Limitations:**

Yes

**Quality:**

2

**Strengths And Weaknesses:**

Strengths

1. [Originality] To my knowledge, this paper is the first to introduce the problem of vector database watermarking and thus this works stands out for blazing a trail and addressing an original problem. The authors successfully position their work as an evolution of prior literature on watermarking for datasets and relational databases and they introduce a corresponding new sets of evaluation metrics to measure watermarking impact on queries in this domain.

2. [Significance] This paper introduces a potentially high-impact problem and presents the first solutions to tackle this challenge. Given the impact that the first relational database watermarking papers have had in both industry and in follow-up academic work, I see this paper has having the potential to be a pioneering work in establishing a new research area.


Weaknesses

1. [Clarity] While I am enthusiastic about the potential for this paper to establish a new, high-impact research direction, I feel the execution in the current version of the paper can be improved considerably. First and foremost, the paper lacks clarity in several places. Perhaps most crucially, the threat model introduced in Section 3.3 is vague and lacks precise definitions. Moreover, while the components of a watermark are described in Section 4.1, I don't believe there is a formal definition of watermarking included in the paper. Beyond just orienting readers to the nature of the problem, I think these formal definitions are critical to be able to judge the quality of the proposed watermarking schemes presented in the paper. For example, in Section 3.3, the authors mention that their threat model assumes attackers are "not too strong" but it is unclear what exactly this means. Is the attacker computationally bounded in some manner (e.g. it can run only in polynomial time)? What would then be the time complexity of an attacker trying to recover the watermark? I think this paper would benefit greatly from a revision that introduces formal definitions of the key concepts behind a watermark and then organizes the evaluation of the paper in relation to these definitions. I believe this change would improve the clarity of this work quite a bit.

2. [Quality] In addition to improving the quality of the paper's presentation, I think the authors could also make this submission stronger by either improving the efficacy of their proposed watermarking schemes, or presenting a more comprehensive case that their schemes are viable solutions to the vector DB watermarking problem. For example, the authors present random selection (RS) as a universal method that can be applied to any vector DB regardless of how the index is implemented. But Table 2 seems to suggest that a tabular watermark can match or even exceed the performance of RS. It would be interesting to understand why this is the case. Does TabularMark work as well as RS on other datasets beyond SIFT 1M or is this a one-off result. Moreover, I think it is critical for the authors to evaluate their technique on additional vector datasets, especially those with embeddings derived from modern deep learning models (SIFT is not a deep learning based dataset). Furthermore, while TVP attains the best performance when compared to the baseline methods in Table 2, it is unclear if result this generalizes to other datasets, or even if the transparent vector technique proposed by the authors extends to other datasets. I think this is a critical question to address. Another weakness of TVP is that the technique is presented as an approach highly specialized to HNSW, which limits its impact. Could TVP work for other navigable small world graph indexes like Vamana/DiskANN or NSG? This would give me more confidence in the generality of the technique. With a more comprehensive experimental evaluation, I believe this paper can be much stronger.

---

> ### Author Rebuttal · Authors · 2025-07-29
>
> Dear Reviewer Ncy6:
>
> We sincerely appreciate the insightful and constructive feedback provided by the reviewers, which has been crucial in helping us improve the quality of our work. We are particularly grateful for your recognition of the originality and potential impact of this paper. As you pointed out, this work presents the first dedicated scheme and evaluation metrics, laying the foundational framework for vector database watermarking, which may open up a new research direction.
>
> In response to your valuable suggestions, we have made the following revisions: we have provided formal definitions of watermarks and threat models (see responses to Questions 1 and 2), and expanded the experimental evaluation to two additional large-scale datasets to further demonstrate the generality and effectiveness of our method (see response to Question 3 and the newly added Tables 1-2).
>
> We hope these modifications adequately address your concerns. We sincerely hope you will consider increasing your score, as your support is crucial for advancing this research direction.
>
> ## **W1 & Q1.**
> > **Can you provide a precise definition of a watermark?**
>
> Thank you for raising this important question. Below, we provide the **formal definition of watermarks in vector databases**. We will incorporate this definition into the revised manuscript if deemed appropriate.
>
> ### **1. Watermarking Algorithms**
>
> The watermark $W \in \\{0, 1\\}^L$ is a binary string that can be embedded into digital carriers for copyright verification.
> $\mathcal{D} = \\{V^1, V^2, \cdots, V^n\\}$ is a vector database containing $n$ $d$-dimensional vectors.
>  A watermarking scheme for vector database $\mathcal{D}$ consists of the following two algorithms:
>
> * **Embedding Algorithm**:  \$\mathsf{Em}(\mathcal{D}, W, \theta\_1) \rightarrow \mathcal{D}\_w\$
> **Input**: the original database \$\mathcal{D}\$, the watermark \$W\$, and the embedding parameter $\theta_1$ (which includes encryption parameters and adjustment parameters for controlling embedding impact and robustness).
> **Output**: the watermarked database \$\mathcal{D}\_w\$.
>
> * **Extraction Algorithm**:  \$\mathsf{Ex}(\mathcal{D'}, \theta\_2) \rightarrow W’\$
> **Input**: a database \$\mathcal{D'}\$ that may have been modified and extraction parameters \$\theta\_2\$ (which includes decryption parameters and recovery parameters for ensuring accurate watermark extraction).
> **Output**: the extracted watermark \$W’\$.
>
> Watermarking schemes have two major pursuits: low embedding impact and high robustness. The evaluation criteria of these two properties for vector database watermarking are presented below.
>
> ### **2. Embedding Impact**
>
> The embedding algorithm $\mathsf{Em}(\cdot)$ selects a set of vectors—called the carrier vector set $\mathbf{C_V} \subseteq \mathcal{D}$ —whose elements may be modified to encode the watermark.
> These modifications may affect ANN queries. To quantify this impact, we introduce two metrics (detailed definitions in sections 5 and 7) :
>
> **Average Missed Query (AMQ)**: Number of true neighbors lost after embedding.
> **Average False Query (AFQ)**: Number of incorrect neighbors newly introduced after embedding.
>
> Lower AMQ and AFQ values indicate smaller embedding impact and higher transparency.
>
> ### **3. Robustness**
>
> After obtaining the watermarked database $\mathcal{D}\_w\$, an attacker may attempt to remove the watermark by modifying the data, thereby tampering with the database to $\mathcal{D}'\$, which may cause errors in watermark extraction. The robustness of the watermark can be evaluated using the bit error rate (BER):
>
> $$
> \text{BER} \triangleq \frac{\sum_{i=1}^L W_i \oplus W_i'}{L}
> $$
>
> The lower the BER, the higher the robustness against adversarial interference.
>
> Our proposed TVP scheme fits within this framework: it selects carrier vectors with minimal query impact , thereby minimizing AMQ and AFQ. Simultaneously, it ensures robust extraction under common perturbations, achieving low BER.
>
>
> ## **W1 & Q2.**
> > **Define threat model and explain the meaning of "not too strong" in Section 3.3.**
>
> We appreciate your interest in the threat model and would like to clarify its definition here.
>
> The core threat we focus on is watermark removal attacks, where attackers attempt to destroy watermarks by deleting or disturbing part of the vector, thereby evading ownership verification. The essence of such attacks is not to crack a certain encryption mechanism, and their effectiveness does not depend on the attacker's computing resources.  Therefore, the statement in the threat model that “**the attacks are usually not too strong**” does not refer to limitations on the attacker's **computational power**, but rather to limitations on the **intensity** of the attacker's attacks.
>
> Attackers are also users of the vector database, so while attempting to remove the watermark, they must also maintain the database's availability. If overly intense attack methods are employed, it may result in severe degradation of approximate nearest neighbor (ANN) query results, rendering the database virtually unusable.
>
> We have demonstrated the relationship between attack intensity and query errors in Appendix D (Figure 9): when the proportion of vectors deleted or altered by attackers reaches 50%, query errors also approach 50%. This indicates that once attackers exceed a certain modification threshold, the database's functionality becomes unsustainable. Therefore, we reasonably assume that attackers will not employ overly strong attacks, forming the fundamental premise of our threat model.
>
> To eliminate ambiguity, we will revise the description of the threat model in the paper. Specifically, we will clarify that “not too strong” refers to attack intensity restrictions, rather than computing power restrictions. Based on our experimental results, we will also suggest a rough upper bound for attack intensity—for example, modifying more than 30% of vectors may already lead to unacceptable query degradation. This revision will enhance the rigor and clarity of our threat assumptions.
>
> ## **W2 & Q3.**
> > **Experiments on other datasets.**
>
> We thank you for your valuable suggestions regarding the generalization of the experiments.
>
> Since “transparent” vectors (i.e., vectors with relatively low query frequencies) are common in any dataset, we speculate that the TVP method has good cross-dataset generalization capabilities.
>
> To validate this hypothesis, we conducted extended experiments on two widely used deep learning embedding datasets: arxiv-nomic-768-normalized (derived from VIBE, representing text summaries) and deep1b (a classic large-scale image vector dataset). The experimental parameters were consistent with those in the paper. We evaluated the average missed queries (**AMQ**) and average false queries (**AFQ**) caused by different watermark embedding schemes, with experimental results shown in Tables 1 and 2.
>
>
> Table 1: Experimental results on the Deep1b dataset
>
> | Scheme | SCPW | TabularMark | RS | TVP |
>  |------------|------|-------------|-----|------|
>  | **AMQ** | 13.96| 13.38 | 13.42| **3.17** |
> | **AFQ** | 14.21| 13.73 | 13.59| **3.29** |
>
> Table 2: Experimental results on the arxiv-nomic-768-normalized dataset
> | Scheme | SCPW | TabularMark | RS | TVP |
> |------------|------|-------------|-----|------|
> | **AMQ** | 13.51| 13.83 | 13.75| **4.26** |
> | **AFQ** | 14.17| 14.02 | 14.11| **4.74** |
>
> It can be observed that, on both datasets, the TVP scheme achieves significantly lower average missed query counts (AMQ) and average false positive query counts (AFQ) compared to other methods. This indicates that the strategy of prioritizing “transparent” vectors for modification can also reduce query errors on other datasets, thereby validating the cross-dataset generalization capability of our method.
>
> Additionally, we note that on both datasets, TabularMark performs very similarly to the random selection (RS) method, with little difference between the two in terms of AMQ and AFQ.
>
> ## Weakness 2.
>
> > **Is the TVP scheme applicable to other ANN indexes?**
>
> We believe that the Transparent Vector Priority (TVP) concept has good generalizability, not only applicable to HNSW but also extendable to other mainstream ANN index structures (such as IVF, NSG, LSH, etc.), simply by redefining the transparency parameters according to the specific characteristics of the index.
>
> To verify the universality of transparent vectors, we tested the frequency distribution of vector queries across multiple index structures and used the Gini coefficient to measure their imbalance. A higher Gini coefficient indicates a more uneven query frequency distribution. The experimental results are shown in Table 3.
>
> Table 3: Gini coefficient of vector query frequency under different ANN index structures.
>
> | Index | HNSW | IVFPQ | NSG | IVFFlat | LSH |
> |----------------|--------|--------|--------|---------|--------|
> | **Gini coefficient** | 0.3558 | 0.3683 | 0.4203 | 0.4132 | 0.6097 |
>
> As shown in the table above, the Gini coefficients for other index structures are all higher than those for HNSW, indicating that the distribution of vector query frequencies is more uneven in these structures. This confirms that the transparent vector phenomenon is widely present in various index structures and not limited to HNSW.
>
> Therefore, the “transparent vector priority” approach can be adopted in different index structures to minimize query errors caused by watermark embedding.
>
> This aligns with reviewer ieq6's perspective: “Although the paper states that the method is limited to HNSW, it appears to be applicable to other indexing structures as long as the method for determining the transparency parameter is modified.”
> In the future, we will further explore how to design appropriate transparency parameters for other index structures, thereby extending the TVP concept to a broader range of index structures.

---

> > ### Comment · Reviewer_Ncy6 · 2025-08-04
> >
> > Thank you to the authors for their detailed rebuttal. I really appreciate the formal definitions for vector DB watermarking schemes and for clarifying my questions about the threat model. I hope that the authors can incorporate both of these additions into the next revision of the paper.
> >
> > One additional question I have, based on reading the discussion with other reviewers, is the worst-case behavior of the AMQ and AFQ metrics. Reviewer ieq6 raised some good points about the fact that the AMQ and AFQ definitions are query dependent, and pointed out that it reported statistics like variance and considering the impact on top-1 result performance might be important. As an extension of these questions, I wonder if AMQ and AFQ can be made arbitrarily large by selecting the queries in an adversarial manner? More generally, how would you address the concerns that these metrics seem to be sensitive to the choice of query sets?
> >
> > Furthermore, I don't think the authors have addressed my question about why TabularMark and Random Sampling perform about the same. If RS achieves the roughly same performance as a technique not designed for the vector DB case, is it even worth considering as a baseline?
> >
> > Overall, I am excited about this work, but I think the crucial missing piece is still a lack of more comprehensive evaluations on additional datasets and ANN indexes. I appreciate that the authors have addressed this concern in part during the rebuttal period, but I do not think the paper should be accepted without a more thorough evaluation to validate the generality of the proposed techniques. Thus, I will keep my current rating though I again want to emphasize that I see a lot of potential in this work.

---

> > > ### Author Response · Authors · 2025-08-05
> > >
> > > Thank you for your careful review of our work. Upon receiving your latest comments, we immediately began conducting additional experiments to more fully address your concerns. The experiments are currently underway, and we will submit an updated response as soon as they are completed. Thank you for your understanding and patience.

---

> > > ### Author Response · Authors · 2025-08-06
> > >
> > > Thank you for your hard work during the review process. We are very pleased that our clarifications on the watermark definition and threat model have addressed your concerns. As we did in the first rebuttal, we will continue to respond to all your questions with the utmost respect and seriousness.
> > >
> > > ## **Q1**
> > > > **As an extension of these questions, I wonder if AMQ and AFQ can be made arbitrarily large by selecting the queries in an adversarial manner? More generally, how would you address the concerns that these metrics seem to be sensitive to the choice of query sets?**
> > >
> > > Thank you for pointing out the potential sensitivity of AMQ and AFQ to query sets. To address your concern, we conducted additional experiments under varying query compositions. Specifically, we constructed the following four query sets:
> > >
> > > **QS1 (Normal)**: Vectors randomly sampled from a vector database.
> > >
> > > **QS2 (50% Adversarial)**: Half of the vectors are from modified transparent vectors.
> > >
> > > **QS3 (75% Adversarial)**: 75% from modified transparent vectors.
> > >
> > > **QS4 (100% Adversarial)**: All from modified transparent vectors.
> > >
> > > We used these four query sets to test the average missed queries (AMQ) and average false positive queries (AFQ) under the TVP scheme. The experimental results are shown in Table 1.
> > >
> > > Table 1: AMQ and AFQ under TVP for different query sets.
> > >
> > > | Query set                   | AMQ  | AFQ  |
> > > |-----------------------------|------|------|
> > > | QS1  | 3.10 | 3.24 |
> > > | QS2  | 4.19 | 4.47 |
> > > | QS3  | 6.33 | 6.71 |
> > > | QS4  | 8.16 | 8.45 |
> > >
> > > As can be seen, even in the most extreme QS4 scenario (where all vectors in the query set are modified transparent vectors), the AMQ and AFQ values do not exceed 8.5, far better than the baseline (~13). This shows TVP retains low impact even under adversarial queries.
> > >
> > > More importantly, query sets in real-world applications are usually consistent with vector database distributions, similar to the situation in QS1. Adversarial queries such as QS2–QS4 rarely occur in practice, and we conducted these tests solely to address your concerns. We hope that this additional experiment will alleviate your concerns.
> > >
> > > ## **Q2**
> > > > **Furthermore, I don't think the authors have addressed my question about why TabularMark and RS perform about the same. If RS achieves the roughly same performance as a technique not designed for the vector DB case, is it even worth considering as a baseline?**
> > >
> > > TabularMark does not incorporate optimization mechanisms to reduce embedding impact, so its AMQ and AFQ are roughly equivalent to those of RS, the similarly unoptimized baseline scheme.
> > >
> > > Moreover, TabularMark requires comparing the original data with the watermarked data during watermark extraction. This requires copyright owners to store two copies of the data, resulting in additional storage overhead, which is unacceptable for large vector databases. Therefore, TabularMark is not suitable for vector databases. To address this issue, we propose RS, which enables watermark extraction without requiring the original data, making it more suitable for vector databases. We will explain the limitations of TabularMark in this article to avoid misunderstandings.
> > >
> > > It is worth noting that RS achieves performance comparable to TabularMark under weaker extraction conditions, highlighting its advantages.
> > >
> > > ## **Q3**
> > > > **Overall, I am excited about this work, but I think the crucial missing piece is still a lack of more comprehensive evaluations on additional datasets and ANN indexes. I appreciate that the authors have addressed this concern in part during the rebuttal period, but I do not think the paper should be accepted without a more thorough evaluation to validate the generality of the proposed techniques.**
> > >
> > > Thank you for your interest in the generalizability of TVP. We have already supplemented the relevant experiments in our previous rebuttal, and the results show that the embedding impact of TVP is lower than the baseline on two new datasets. We have also verified the existence of transparent vectors in multiple mainstream ANN indexes, so other indexes can also utilize transparent vectors to reduce embedding impact. We believe that these empirical results sufficiently demonstrate the generalizability of TVP across datasets and index structures.
> > >
> > > We appreciate your acknowledgment of the efforts we have made in verifying generalizability. You also mentioned the lack of “more thorough evaluation.” We are eager to fully address your concerns, but it is currently unclear what specific supplementary experiments you expect to be included in a “more thorough evaluation.” If we could receive more explicit guidance, we would be more than happy to supplement the relevant experiments as soon as possible.

---

> > > > ### Comment · Reviewer_Ncy6 · 2025-08-06
> > > >
> > > > Thank you to the authors for their continued engagement and efforts during the discussion period. I have found the authors' clarifications to be very helpful and I hope they can incorporate these points into the next revision of the paper. In particular, I think the points about the behavior of TVP under adversarial query sets and the discussion of the limitations of TabularMark are very helpful and answer those questions I raised.
> > > >
> > > > Regarding a "more thorough evaluation" what I mean is computing the AMQ and AFQ metrics on ANN indexes beyond HNSW such as NSG, IVF, LSH, etc. I really appreciate the authors computing Gini coefficients to argue for the existence of transparent vectors in other indexes, but I think this evidence on its own is insufficient and it is essential for completeness to compute the actual watermark metrics on these indexes. I think this is critical because either 1) the TVP technique generalizes as expected which confirms the broad applicability of the technique and rules out any surprises from moving to other indexes or 2) TVP doesn't work as well on these other indexes which raises interesting questions as well. Either way, I think there is something to learn here to complete the story presented in the paper.
> > > >
> > > > I think the authors' experiments on the two additional datasets in the rebuttal period is also a fantastic addition. I think it would be valuable to evaluate on even more datasets (such as from VIBE or the large-scale Big ANN benchmarks) but I recognize this may not be feasible to complete during the rebuttal period and this is not as critical as my concern above about claiming TVP generalizes to other indexes without hard evidence.

---

> > > > > ### Author Response · Authors · 2025-08-06
> > > > >
> > > > > Thank you very much for your continued participation and valuable feedback. We have carefully read your latest comments and are working hard to prepare a detailed response to address your concerns. Thank you for your patience.

---

> > > > > ### Author Response · Authors · 2025-08-08
> > > > >
> > > > > We greatly appreciate your more specific feedback and are pleased that the discussion regarding the behavior of TVP under adversarial query sets and the limitations of TabularMark has addressed your concerns. We thank you for acknowledging our cross-dataset testing experiments. As an additional clarification, the new dataset we introduced, arxiv-nomic-768-normalized, is derived from the VIBE you mentioned.
> > > > >
> > > > > Regarding your main concern about “calculating AMQ and AFQ metrics on ANN indexes other than HNSW,” we have supplemented the experiments with four ANN index structures: NSG, LSH, IVFFlat, and IVFPQ. We counted the query frequencies of all vectors in the index structure and prioritized embedding watermarks (TVP) in vectors with low query frequencies, while using randomly selected vectors for watermark embedding as a comparison scheme (RS). We then calculated the embedding impact (AMQ/AFQ) for each case. The experimental results are as follows:
> > > > >
> > > > > | Index Type | TVP (AMQ / AFQ) | RS (AMQ / AFQ) |
> > > > > |---------|------------------|------------------|
> > > > > | NSG | 0.12 / 0.09 | 0.38 / 0.27 |
> > > > > | LSH | 0.75 / 0.95 | 2.27 / 2.49 |
> > > > > | IVFFlat | 1.46 / 1.52 | 3.57 / 3.81 |
> > > > > | IVFPQ | 6.46 / 9.34 | 17.38 / 19.21 |
> > > > >
> > > > > As can be seen, regardless of the index type, TVP consistently outperforms RS, directly validating its universality across indexing scenarios.
> > > > >
> > > > >
> > > > > Finally, we would like to reiterate the main contributions of this paper.
> > > > >
> > > > > **RS**: Proposed the first universal vector database watermarking scheme, RS;
> > > > > **AMQ and AFQ**: Proposed the AMQ and AFQ metrics for measuring embedding impact, establishing a systematic evaluation framework for the field;
> > > > > **TVP**: A TVP optimization method was proposed, which significantly reduced the embedding impact by prioritizing the modification of transparent vectors with low query frequencies.
> > > > >
> > > > > We hope to receive your support to enable the academic community to access this research earlier and drive the development of this field.

---

> > > > > > ### Comment · Reviewer_Ncy6 · 2025-08-08
> > > > > >
> > > > > > Thank you again to the authors for their continued engagement and efforts during the discussion period. I am very pleased to see the additional results on other ANN index types as I think this is a critical experiment to validate the core contributions of the paper. The new results also look quite strong.
> > > > > >
> > > > > > Based on the authors' latest response, I will raise my score to a 4 and will advocate for acceptance of this paper. I choose not to assign a higher score at this time because I think the paper would still benefit from a more comprehensive evaluation on more benchmark datasets (at least 5-6 in total), and I hope the authors can consider adding these additional results to the camera-ready version of the paper.
> > > > > >
> > > > > > I congratulate the authors on their strong efforts to address reviewer concerns during the rebuttal and discussion period. I think their proposed additions to the paper (formal definitions of watermarking and the threat model, discussions around the adversarial robustness of their AMQ and AFQ metrics, and more thorough experiments on additional datasets and indexes) make the paper much, much stronger than their initial submission. Thus, I am happy to recommend acceptance of this paper.

---

> > > > > > > ### Author Response · Authors · 2025-08-08
> > > > > > >
> > > > > > > We sincerely appreciate your constructive and insightful review comments. Your guidance during the discussion has greatly helped us refine the paper, enabling us to improve the quality of our work through clearer explanations, more rigorous formal definitions, and richer experiments. We are pleased that these improvements have significantly enhanced the completeness and persuasiveness of the paper, and we will incorporate your suggested additions into the final camera-ready version.

---

### Official Review · Reviewer_7c4a · 2025-07-06

**Clarity:** 2
**Significance:** 2
**Originality:** 2
**Rating:** 4
**Confidence:** 3

**Summary:**

The paper introduces Transparent Vector Priority (TVP) watermarking, a novel scheme designed for embedding watermarks into vector databases, specifically focusing on minimizing the impact on Approximate Nearest Neighbor (ANN) query results. Vector databases, which are crucial for machine learning tasks, are susceptible to security threats like unauthorized replication, and watermarking offers a solution for ownership authentication.

The core challenge addressed is that modifying vectors to embed a watermark can negatively affect ANN query performance. The authors observe that within widely used indexing structures like Hierarchical Navigable Small World (HNSW), some "transparent" vectors have significantly fewer edges or no edges at all and consequently exhibit much lower query frequencies.

Key Contributions of this work are:

-C1. Novel Watermarking Scheme: The paper proposes the Transparent Vector Priority (TVP) watermarking scheme specifically for vector databases, addressing the critical security need for ownership authentication.

-C2. Minimizing Query Impact: It introduces a unique approach by identifying and prioritizing "transparent" vectors (low-query-frequency vectors with fewer or no edges in HNSW graphs) for watermark embedding, thereby minimizing the degradation of ANN query results.

**Questions:**

Q1- The proposed TVP watermarking scheme effectively leverages the characteristics of "transparent" vectors within the HNSW indexing structure. Have you explored, or do you have plans to investigate, how this scheme might be adapted or optimized for other widely used ANN indexing structures (e.g., IVF, tree-based methods, or different graph-based approaches) that may not exhibit the same 'transparent' vector properties as HNSW?

Q2- Beyond the HNSW structure, how might the concept of prioritising low-impact vectors be identified and applied in other types of vector database architectures or even different data management systems where modifications are necessary but impact needs to be minimised?

**Ethical Concerns:**

["NO or VERY MINOR ethics concerns only"]

**Final Justification:**

The paper addresses the novel challenge that embedding watermarks in vectors can degrade approximate nearest neighbor (ANN) query performance. Overall, basic idea is new and has promising strength. It provides new approach for the outstanding problem for good real impacts.
It observes that in popular indexing structures like Hierarchical Navigable Small World (HNSW): some “transparent” vectors, those with few or no edges, are queried much less frequently. Leveraging key innovation, the authors propose the Transparent Vector Priority (TVP) watermarking scheme for vector databases. TVP embeds watermarks preferentially in these low-query-frequency vectors to reduce the impact on ANN query accuracy. This approach aims to balance robust ownership authentication with minimal performance loss.

**Limitations:**

L1- Robustness Against Sophisticated Attacks: The paper focuses on ownership authentication and minimising query impact. While all watermarking schemes face challenges, the specific robustness of TVP against various advanced adversarial attacks (e.g., subtle noise injection, vector perturbations, or targeted attempts to remove the watermark without destroying the data's utility) is not detailed as a primary focus of this work.

L2- Subtle Impact on "Transparent" Vectors: Although "transparent" vectors have low query frequency, modifying them still alters the original data. In highly specialized or niche analytical tasks, even these seldom-queried vectors might hold unique information, and their alteration, however minimal, could theoretically have unforeseen, tiny impacts on very specific data analysis or rare queries.

**Quality:**

2

**Strengths And Weaknesses:**

1. Strengths:

S1- Addresses a Critical Security Need: The paper tackles the important and practical problem of ownership authentication in vector databases, which are increasingly vital for machine learning applications and susceptible to unauthorized replication.

S2- Minimises Impact on Query Performance: A key strength is the innovative approach of identifying and prioritising "transparent" vectors (those with fewer or no edges and low query frequency in HNSW graphs) for watermark embedding. This strategic placement significantly reduces the negative impact on Approximate Nearest Neighbor (ANN) query results, which is often a major drawback of watermarking schemes.

S3- Leverages HNSW Structure Intelligently: The scheme cleverly exploits the inherent characteristics of the widely used Hierarchical Navigable Small World (HNSW) indexing structure, demonstrating a deep understanding of how these databases operate.

2. Weaknesses: Dependence on Specific Indexing Structure- While a strength, the method's reliance on the existence and characteristics of "transparent vectors" derived from HNSW's edge selection and pruning strategies suggests that its direct applicability or optimal performance might be limited to HNSW or very similar graph-based indexing structures. Its generalizability to other types of ANN indexes (e.g., IVF, tree-based, or pure hashing methods) is not explicitly discussed and might be challenging

---

> ### Author Rebuttal · Authors · 2025-07-29
>
> Dear Reviewer 7c4a:
>
> We appreciate your insightful and valuable comments. We have carefully considered each comment and responded to them individually, and sincerely hope that our responses have adequately addressed your concerns. If so, we would be grateful if you could raise your score. If not, please let us know your further concerns, and we will continue to respond positively to your comments and improve our submission.
>
> ## Weakness & Question 1.
>
> > **1. Is the TVP scheme applicable to other ANN indexes?**
>
> We believe that the Transparent Vector Priority (TVP) concept has good generalizability, not only applicable to HNSW but also extendable to other mainstream ANN index structures (such as IVF, NSG, LSH, etc.), simply by redefining the transparency parameters according to the specific characteristics of the index.
>
> To verify the universality of transparent vectors, we tested the frequency distribution of vector queries across multiple index structures and used the Gini coefficient to measure their imbalance. A higher Gini coefficient indicates a more uneven query frequency distribution. The experimental results are shown in Table 1.
>
> **Table 1**: Gini coefficient of vector query frequency under different ANN index structures.
>
> | Index | HNSW | IVFPQ | NSG | IVFFlat | LSH |
> |----------------|--------|--------|--------|---------|--------|
> | **Gini coefficient** | 0.3558 | 0.3683 | 0.4203 | 0.4132 | 0.6097 |
>
> As shown in the table above, the Gini coefficients for other index structures are all higher than those for HNSW, indicating that the distribution of vector query frequencies is more uneven in these structures. This confirms that the transparent vector phenomenon is widely present in various index structures and not limited to HNSW.
>
> Therefore, the “transparent vector priority” approach can be adopted in different index structures to minimize query errors caused by watermark embedding.
>
> This aligns with reviewer ieq6's perspective: “Although the paper states that the method is limited to HNSW, it appears to be applicable to other indexing structures as long as the method for determining the transparency parameter is modified.”
> In the future, we will further explore how to design appropriate transparency parameters for other index structures, thereby extending the TVP concept to a broader range of index structures.
>
> ## Question 2.
> > **2. Methods for identifying low-impact vectors and the potential of TVP concepts in different data management systems.**
>
> In any vector database, a direct method for identifying low-impact (i.e., “transparent”) vectors is to count their query frequencies. However, this method incurs excessive computational overhead in large-scale systems. To efficiently identify low-impact vectors, we need to deeply understand the internal mechanisms of various index structures. For example, in HNSW, edge nodes have lower query frequencies, while in PQ, quantization errors and cluster density may influence query probabilities.
>
> Based on these structural characteristics, we can design more efficient heuristic strategies to estimate vector impact without enumerating queries. This is a challenging yet worthwhile direction for further research.
>
> More broadly, the TVP approach can be viewed as a general data perturbation strategy applicable to other data management systems: prioritizing the modification of low-impact data without significantly affecting critical system performance (such as accuracy or availability). To systematically promote this strategy, we propose a general three-step framework:
>
> 1.**Impact Quantification**: Define system-specific impact metrics (e.g., query frequency, access heat, node centrality);
> 2.**Low-Impact Target Identification**: Combine statistical or structural heuristics to identify low-impact data units;
> 3.**Targeted Modification**: Implement minimized modifications under system constraints to reduce side effects.
>
> We believe this framework has good general applicability and plan to explore its implementation in more data systems in future work.
>
> ## Limitation 1.
>
> > **3. Robustness against sophisticated attacks was not discussed in detail.**
>
> We appreciate your comments on the stability assessment. We have conducted various adaptive attack experiments targeting TVP, including subtle perturbation injection (referred to as “adaptive modification attacks” in the paper), and quantified the extraction performance using BER (bit error rate). Due to space limitations, the results of these experiments are presented in Figure 10 of Appendix D.
>
> We will move these experimental results to the main text in the revised version based on your review comments. Once again, we sincerely appreciate the reviewers' valuable feedback.
>
> ## Limitation 2.
>
> > **4. Modifications to low-frequency but semantically critical vectors may introduce unforeseen minor impacts.**
>
> We appreciate your forward-thinking question, which has helped us further consider the applicability of TVP in extreme scenarios.
>
> We fully acknowledge that in certain specific application scenarios, modifying vectors with low query frequencies that possess unique semantic meanings or carry critical information may result in minor but unforeseen impacts.
>
> However, this issue is not unique to TVP but rather a common challenge shared by most data perturbation-based watermarking schemes. There is inherently a trade-off between security and usability.
>
> To address this issue, this paper not only proposes the “Transparent Vector Priority” concept to minimize impact but also provides adjustable parameters for the extent of data modification to further control potential distortion.
>
> In the future, we plan to integrate multi-dimensional information such as semantic labels and task sensitivity to further identify vectors that are “low-impact” in both the “access frequency” and “semantic importance” dimensions, thereby enhancing the controllability of TVP.

---

### Comment · Area_Chair_F2wU · 2025-08-04

Dear Reviewers,
Thanks for your efforts. Since authors have replied to the reviews, please check whether the rebuttal solves your concerns and respond to authors.

Best regards,
AC

---

### Note · Authors · 2025-08-12

**Dear Area Chair and Reviewers,**

We sincerely thank the AC and reviewers for their efforts.

This paper presents the first watermarking scheme specifically designed for vector databases, defines the AMQ and AFQ metrics to measure query errors introduced by watermark embedding, and proposes the Transparent Vector Priority (TVP) strategy, which reduces query errors by approximately 75%.
These contributions have been widely recognized by reviewers.

**Key Positive Feedback:**

- Reviewer 7c4a: _"Addresses a Critical Security Need" and "Novel Watermarking Scheme"_

- Reviewer Ncy6: _"Potential to be a pioneering work in establishing a new research area"_

- Reviewer ir5J: _"Fills a gap in the research field" and "shows a smart optimization approach"_

- Reviewer ieq6: _"Appears to be robust against deletion attacks" and "Appears to be applicable to other indexing structures"_

- Reviewer SNQH: _"Innovatively uses HNSW's graph properties to embed watermarks"_

During the rebuttal and discussion phases, we validated the generalizability of TVP across datasets and indexes, evaluated its performance under different parameters and metrics, provided formal definitions of watermarking, clarified questions, and discussed potential future research directions.

Through the rebuttal process, we have addressed all core concerns and gained stronger support from the reviewers:

**Explicit Score Improvements:**
- Reviewer Ncy6: _"Based on the authors' latest response, I will raise my score to a 4 and will advocate for acceptance of this paper."_
- Reviewer ir5J: _"I have decided to raise my score from 3 to 4."_
- Reviewer ieq6: _"Based on these improvements, I have raised my score from 3 to 4."_

The other two reviewers maintained their positive evaluations, so after the rebuttal, all reviewers were inclined to accept the paper.

 We appreciate NeurIPS for facilitating a constructive discussion process.

Sincerely,

Authors

---

### Decision · Program_Chairs · 2025-09-17

**Decision:**

Accept (poster)

**Comment:**

Considering the potential threat to vector dataset, the authors present a Transparent Vector Priority (TVP) watermarking scheme.
All the FIVE reviewers basically appreciate the contributions of this work, and are mostly satisfied with authors' responses:
Reviewer 7c4a:
Novel Watermarking Scheme: The paper proposes the Transparent Vector Priority (TVP) watermarking scheme specifically for vector databases, addressing the critical security need for ownership authentication.
Reviewer: Ncy6
... their proposed additions to the paper (formal definitions of watermarking and the threat model, discussions around the adversarial robustness of their AMQ and AFQ metrics, and more thorough experiments on additional datasets and indexes) make the paper much, much stronger than their initial submission. ... believe the idea of vector database watermarking is a new and timely contribution to the community and has the potential to inspire additional research in this direction.
Reviewer ir5:
...the first kind vector database watermarking scheme (RS) fills a gap in the research field.
Defining missed queries and false queries to measure the impact of watermark embedding on ANN queries is a precise and necessary step. The TVP scheme shows a smart optimization approach.
Reviewer ieq6:
remaining concern: potential serious implications of bit flipping in a vector quantization (VQ) environment.
Reviewer SNQH: satisfied with the authors' rebuttal

Based on the above, the AC suggests to accept this submission.